**Brief Communication**

# Structural insights into the regulation of Cas7-11 by TPR-CHAT

Babatunde Ekundayo[1], Davide Torre [1], Bertrand Beckert[2], Sergey Nazarov[2], Alexander Myasnikov [2], Henning Stahlberg [1]✉ & Dongchun Ni [1]✉

The CRISPR-guided caspase (Craspase) complex is an assembly of the target-specific RNA nuclease known as Cas7-11 bound to CRISPR RNA (crRNA) and an ancillary protein known as TPR-CHAT (tetratricopeptide repeats (TPR) fused with a CHAT domain). The Craspase complex holds promise as a tool for gene therapy and biomedical research, but its regulation is poorly understood. TPR-CHAT regulates Cas7-11 nuclease activity via an unknown mechanism. In the present study, we use cryoelectron microscopy to determine structures of the *Desulfonema magnum* (*Dm*) Craspase complex to gain mechanistic insights into its regulation. We show that *Dm*TPR-CHAT stabilizes crRNA-bound *Dm*Cas7-11 in a closed conformation via a network of interactions mediated by the *Dm*TPR-CHAT N-terminal domain, the *Dm*Cas7-11 insertion finger and Cas11-like domain, resulting in reduced target RNA accessibility and cleavage.

CRISPR (clustered regularly interspaced short palindromic repeats)–Cas systems provide adaptive immunity for host prokaryotes via sequence-directed nucleic acid cleavage and are powerful genetic tools in biomedical research and gene therapy[1,2]. The newly discovered Cas7-11 system uses its bound crRNA as a guide to facilitate highly sequence-specific cleavage of target RNA at two sites separated by six nucleotides[3]. The discovery of this system provides a new tool for sequence-specific RNA targeting for knockdown and editing purposes with remarkably negligible nontarget effects and low cell toxicity[3,4]. CrRNA-bound Cas7-11 assembles into the Craspase complex with an ancillary protein known as TPR-CHAT[5]. TPR-CHAT is a caspase-related protein that regulates Cas7-11 activity via an unknown mechanism and could be involved in viral immunity in host prokaryotes[3–7]. Given the broad range of potential biomedical and therapeutic applications of the Cas7-11 system, understanding of its regulation is crucial for further development as a biotechnological tool.

To investigate the structural basis of regulation of the Craspase complex, we studied the CRISPR subtype III-E loci from *D. magnum* possessing a similar genetic structure to other described systems[3,5] (Fig. 1a and Extended Data Fig. 1a,b). We purified the *Dm*Cas7-11 protein

in complex with crRNA (*Dm*Cas7-11–crRNA) and confirmed its activity by demonstrating its sequence-specific target RNA cleavage in vitro (Fig. 1b and Extended Data Fig. 1c,d). *Dm*Cas7-11–crRNA is stably associated with *Dm*TPR-CHAT to form the Craspase complex when copurified, as shown for other systems[5] (Extended Data Fig. 2). We used cryoelectron microscopy (cryo-EM) to analyze the Craspase complex, which resulted in two structures, one showing clear density for crRNA and all protein domains of the complex (*Dm*Cas7-11–crRNA and *Dm*TPR-CHAT_full), whereas the second lacks density for the regions of *Dm*TPR-CHAT (*Dm*Cas7-11–crRNA and *Dm*TPR-CHAT_NTD). The maps were resolved to overall resolutions of 3.2 Å (0.32 nm) and 3.0 Å, respectively (Extended Data Figs. 3 and 4, Supplementary Fig. 1, Supplementary Videos 1 and 2 and Table 1).

The obtained maps show the Craspase complex to be composed of an elongated structure resembling a 'seahorse', as also described for the structures of type I and type III CRISPR–Cas systems[8–10] (Fig. 1c–f). The crRNA-bound *Dm*Cas7-11 forms the enzymatic core of the complex stably bound to *Dm*TPR-CHAT which contacts multiple Cas7 repeat domains (Fig. 1c–f). The arrangement of *Dm*Cas7-11 domains starts with the amino-terminal Cas7.1 domain forming a cap

[1]Laboratory of Biological Electron Microscopy, Institute of Physics, School of Basic Sciences, EPFL, and Department of Fundamental Microbiology, Faculty of Biology and Medicine, University of Lausanne, Lausanne, Switzerland. [2]Dubochet Center for Imaging, EPFL, University of Lausanne and University of Geneva, Lausanne, Switzerland. ✉e-mail: henning.stahlberg@epfl.ch; dongchun.ni@epfl.ch

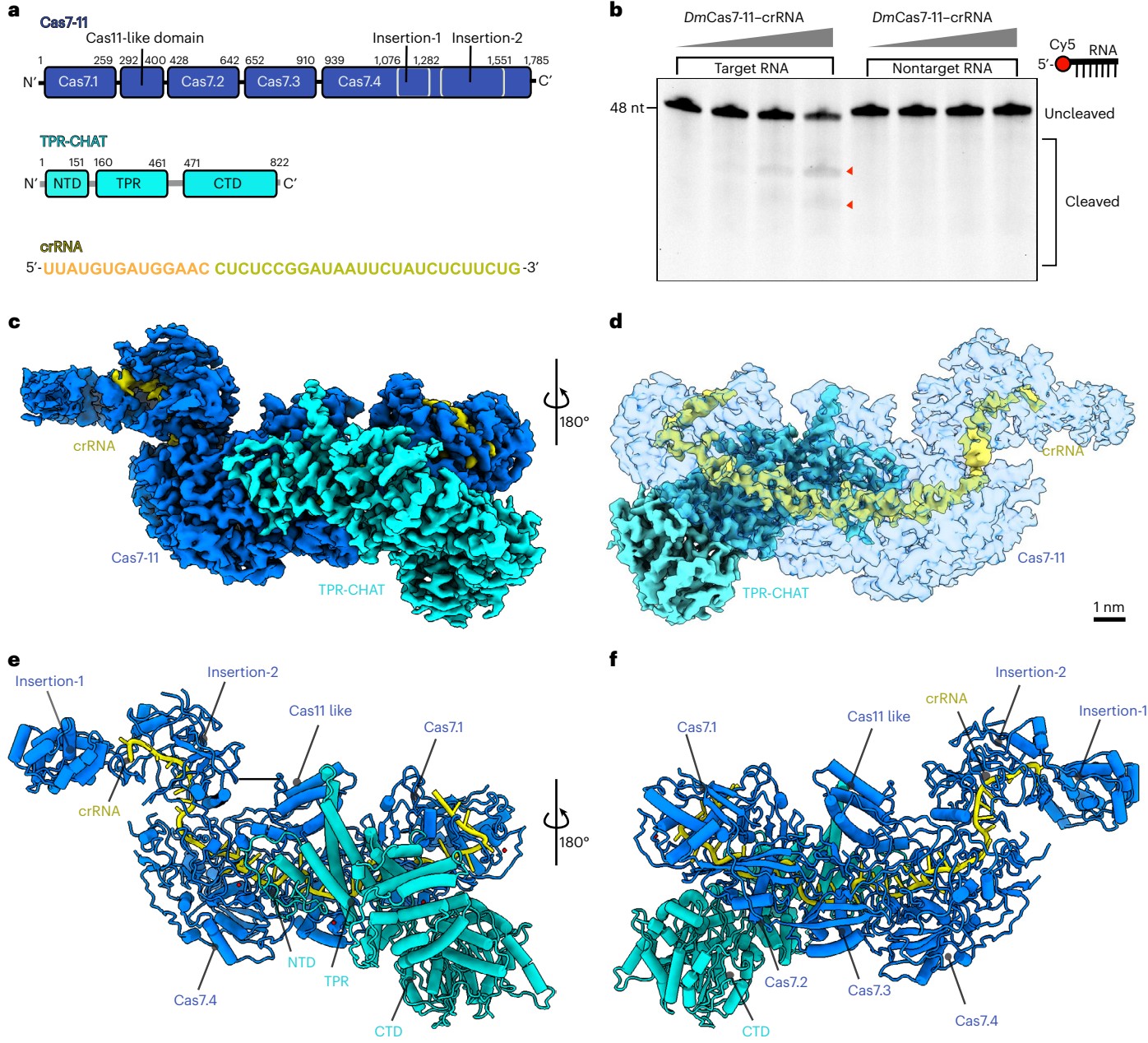

**Fig. 1 | Structure of the *D. magnum* Craspase complex. a**, Domain organization of the Craspase complex. **b**, Denaturing urea–PAGE of *Dm*Cas7-11–crRNA incubated with 48-bp Cy5-labeled target and nontarget RNA. The red arrows indicate the cleavage products (*n* = 5). **c,d**, Cryo-EM density of the Craspase complex in two separate views rotated by 180°. **e,f**, Cartoon representation of the Craspase structure in two separate views rotated by 180°. The protein domains are indicated. Scale bar (**c** and **d**), 1 nm.

at one end of the complex, followed by the interlocking Cas7.2, Cas7.3 and Cas7.4 repeat domains (Fig. 1e–f and Extended Data Fig. 5). The fold of these repeats shares similarities with previously determined Cas7 structures with a 'right-hand' morphology consisting of palm, thumb, web and finger regions[8–10] (Extended Data Fig. 5). However, Cas7.4 is unique, containing a core Cas7 fold with the thumb replaced with the large insertion-1 and -2 subdomains (Fig. 1e–f and Extended Data Fig. 5). The Cas11-like domain (CLD) resides after a loop that extends from the Cas7.1 palm and docks between Cas7.3 and Cas7.4. The CLD forms a protrusion between these repeats and makes multiple interactions with Cas7.2, Cas7.3 and Cas7.4. Another extended loop connects the CLD to the Cas7.2 palm (Fig. 1e–f and Extended Data Fig. 5).

The complex terminates at the other end with insertion-2, followed by insertion-1 of Cas7.4, which forms a 'tail' composed of coiled coils (Fig. 1e–f). Both proteins, insertion-1 and -2, are unique to Cas7.4 and are not present in previously published Cas7 structures from other systems, suggesting that they play a unique role in the function of Cas7-11.

The fully melted crRNA was stably bound to *Dm*Cas7-11, starting from Cas7.1 at its 5′-end, extending toward Cas7.2 and Cas7.3 and terminating in insertion-2 at its 3′-end (Fig. 1d–f and Extended Data Figs. 5 and 6). Given that we could unambiguously assign the nucleotide sequences to the crRNA in our structure, we could confidently determine the positions of the 5′-handle and spacer, revealing the molecular basis of crRNA recognition by *Dm*Cas7-11 (Extended Data Fig. 6). The amino

**Table 1 | Cryo-EM data collection, refinement and validation statistics**

| Data collection and processing | DmCas7-11-TPR-CHAT_full (EMDB-14848; PDB 7ZOQ) | DmCas7-11-TPR-CHAT_NTD (EMDB-14847; PDB 7ZOL) |
|---|---|---|
| Magnification | 96,000x | |
| Voltage (kV) | 300 | |
| Electron exposure (e⁻ Å⁻²) | 60 | |
| Defocus range (−µm) | 0.8–2.5 | |
| Pixel size (Å) | 0.83 | |
| Symmetry imposed | C1 | |
| Initial particle images (no.) | 1,719,283 | |
| Final particle images (no.) | 65,755 | 53,969 |
| Map resolution (Å) | 3.20 | 3.03 |
| FSC threshold | 0.143 | 0.143 |
| **Refinement** | | |
| Initial model used (PDB code) | NA (not applicable) | NA |
| Map sharpening B factor (Å²) | −33.6 | −43.6 |
| Model composition | | |
| Nonhydrogen atoms | 20,972 | 15,444 |
| Protein residues | 2,514 | 1,852 |
| Nucleotides | 44 | 39 |
| Water | 0 | 0 |
| Ligands | ZN:4 | ZN:4 |
| B factors (Å²) | | |
| Protein | 12.7/165.28/58.63 | 0.33/122.55/35.92 |
| Nucleotide | 17.32/264.04/73.71 | 4.91/152.55/22.22 |
| Ligand | 44.52/113.47/71.85 | 18.36/83.28/44.24 |
| R.m.s. deviations | | |
| Bond lengths (Å) | 0.002 (0) | 0.002 (0) |
| Bond angles (°) | 0.530 (4) | 0.552 (2) |
| Validation | | |
| MolProbity score | 1.84 | 1.82 |
| Clash score | 6.69 | 7.02 |
| Poor rotamers (%) | 0.00 | 0.20 |
| Ramachandran plot | | |
| Favored (%) | 92.38 | 93.40 |
| Allowed (%) | 7.38 | 6.27 |
| Disallowed (%) | 0.24 | 0.33 |

acid side chains of Cas7.1 and Cas7.2 recognize the 13 nt of the 5′-handle mainly through sequence-specific interactions. Notably, nucleotides 2–8 of the crRNA 5′-handle adopt a canonical 'hook-like' structure stabilized by interactions involving the three nucleotides U6, G7 and A8 and charged amino acid residues from Cas7.1 (refs. [9,10]) (Extended Data Fig. 6d–f). The crRNA spacer starts from the 5′-nucleotide C15 and terminates at nucleotide G40 at the 3′-end. The spacer divides into two segments that interact with Cas7-11 via nonsequence-specific interactions between the sugar–phosphate backbone of the spacer and amino acid side chains from Cas7.3 and Cas7.4 (Extended Data Fig. 6g–i).

The absence of sequence-specific interactions allows for target RNA recognition by complementary base-pairing with the bound spacer, similar to structures of other type I and III systems[8–10] (Extended Data Fig. 6c). During the manuscript revision of the present study, publications describing the cryo-EM structures of Cas7-11 from another two species were released[11,12]. These structures share overall similarity to the one in the present study, with striking differences in the organization of the insertion domains and the position of the CLD (Extended Data Fig. 7).

DmTPR-CHAT, the second protein component of the Craspase complex, is implicated in regulating or tuning Cas7-11 RNA nuclease activity via an elusive mechanism[3]. Our structure of the Craspase complex reveals the full-length structure of DmTPR-CHAT, the molecular basis of its interaction with Cas7-11 and molecular insights for its regulatory role in the Craspase complex (Figs. 1 and 2 and Extended Data Figs. 8, 9 and 10). DmTPR-CHAT organizes into three domains: the N-terminal domain (NTD), tetratricopeptide repeat (TPR) domain and the carboxy-terminal domain (CTD) (Fig. 1e–f and Extended Data Fig. 8a). The CTD contains a caspase-like subdomain (CLS) starting from β sheet-4 at its N terminus and terminating at β sheet-11 at its C terminus (Extended Data Fig. 8b). The CLS is of particular interest because predictions suggest that it acts as a trigger for cell death or dormancy via a caspase-related peptidase activity[3–7]. Structural alignments of DmTPR-CHAT with other caspases showed high similarity in the CTD and TPR with Human Separase[13–15] (Extended Data Fig. 8c,d). The CLS was also similar to Cas7 (± substrate) and PIGK (phosphotidylinositol glycan anchor biosynthesis class K) with the root mean square deviation (r.m.s.d.) between the C-α atoms of the protein backbones of 1.1 and 1.4 Å (Extended Data Fig. 8e). In addition, we could also identify a putative catalytic cysteine residue (Cys728) and other residues involved in substrate recognition in the CLS from the structural alignments, supporting the predictions[13–15] (Extended Data Fig. 8e,f). These findings show that the CTD harbors a putative caspase-related peptidase, with its substrate yet to be determined.

DmTPR-CHAT makes multi-domain interactions with DmCas7-11−crRNA via its NTD and CTD, forming two major binding sites. The binding site of the NTD has a larger surface area than that of the CTD (Extended Data Fig. 9). The NTD interacts with the Cas7.4 palm, a loop extending from the Cas7.3 palm, and the insertion finger from Cas7.4 via salt bridges and hydrogen bonds (Extended Data Fig. 9a,d). At the other site, the CTD forms salt bridges with residues from the palms of Cas7.1 and Cas7.2 (Extended Data Fig. 9a–c). Cryo-EM data processing of the Craspase complex revealed flexibility of the TPR and CTD, in contrast to the NTD which is rigidly bound to the complex (Fig. 2a, Extended Data Fig. 3 and Supplementary Fig. 2). These findings show that the NTD is sufficient for the association of DmTPR-CHAT with DmCas7-11−crRNA.

The overall structure of D. magnum Craspase revealed that the complex adopts a 'closed' conformation characterized by the steric occlusion of the first segment of the spacer crRNA by the CLD (Fig. 2b). Notably, we observed from two-dimensional (2D) classes of DmCas7-11−crRNA that a feature corresponding to the CLD is not always present (Fig. 2c and Supplementary Fig. 3). This finding reveals that the CLD in DmCas7-11−crRNA can alternate between a rigid and a flexible state facilitated by its location between extended loops that form long linkers flanking the domain (Supplementary Video 3). In contrast, the CLD in the Craspase complex stably associates with the complex, showing that DmTPR-CHAT promotes the rigidity of the CLD (Fig. 2c,e). Asp310 or Asp307 (CLD) and Lys1526 (insertion finger) form a salt bridge facilitated by interactions between the insertion finger and the NTD (Fig. 2b and Supplementary Video 4). This network of interactions stabilizes the CLD. Strikingly, the highly conserved amino acid Asp682 implicated in nuclease activity is proximal to this region, indicating that target cleavage occurs nearby (Fig. 2b and Extended Data Fig. 6g). Published studies using the Cas7-11 system from Scalindua brodae (SbCas7-11) and Desulfonema ishitimonii (DiCas7-11) suggest an unclear role in the association

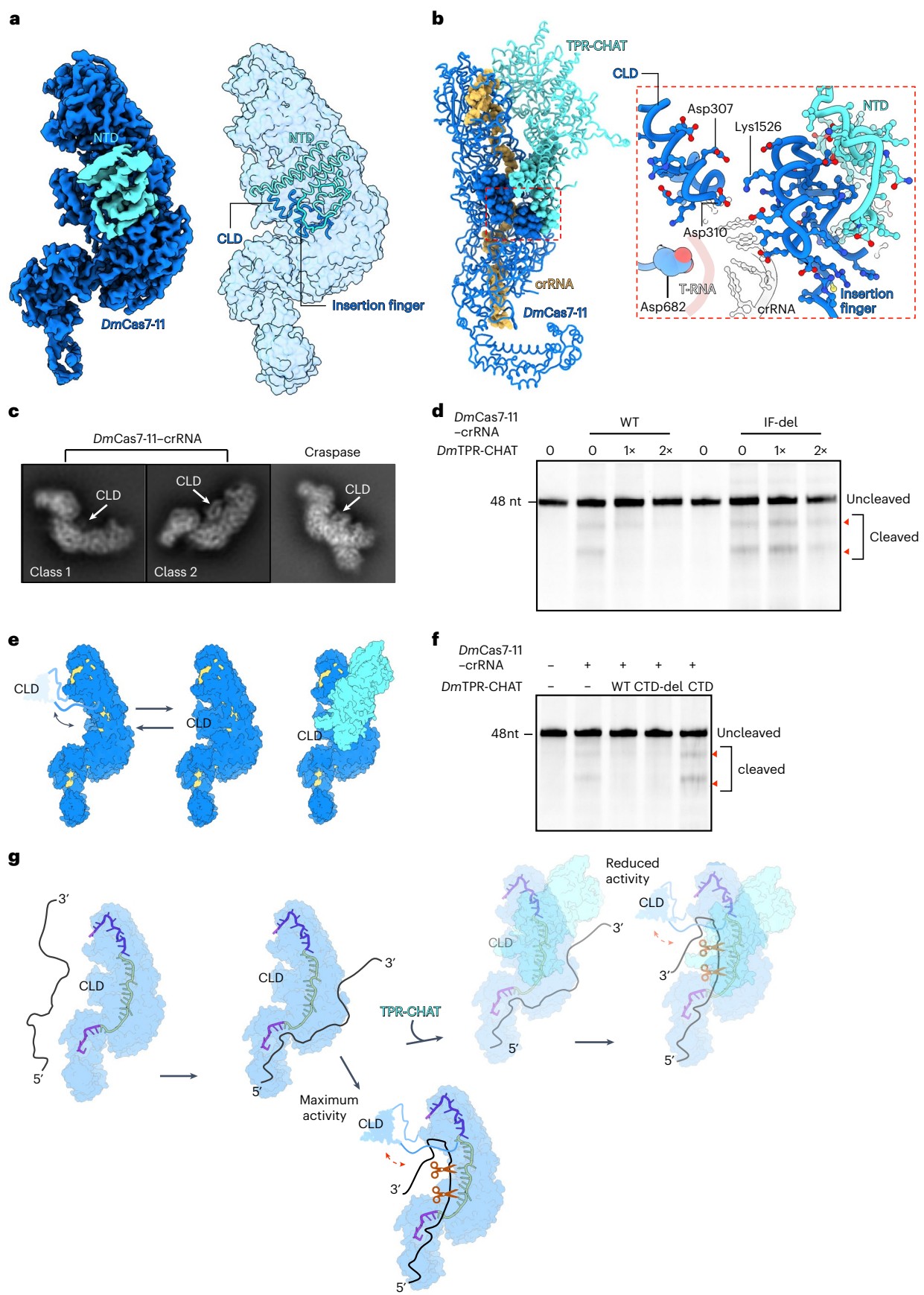

between TPR-CHAT and *Dm*Cas7-11–crRNA[3,5]. The study using the *Di*Cas7-11 showed that TPR-CHAT regulates Cas7-11 by reducing its nuclease activity, whereas a study using the *Sb*Cas7-11 showed that TPR-CHAT did not affect nuclease activity[3–5]. In the present study, we utilize in vitro assays to reveal that *Dm*TPR-CHAT can stably inhibit the nuclease activity of *Dm*Cas7-11 by reducing target RNA binding (Fig. 2d,f

**Fig. 2 | A mechanistic model of *Dm*Cas7-11 regulation by *Dm*TPR-CHAT. a**, Left, the cryo-EM density of the Craspase complex, containing only density for the NTD from *Dm*TPR-CHAT (*Dm*Cas7-11–crRNA–TPR-CHAT$_{NTD}$). Right, the positions of the CLD, NTD and insertion finger. **b**, Left, the 'closed' conformation of the Craspase complex highlighted by the red dashed box. Right, the details of the interactions in the box of CLD, insertion finger and NTD. **c,e**, The 2D classes from *Dm*Cas7-11–crRNA and the Craspase complex, with cartoon representations of these 2D classes shown in **e**. White arrows indicate the positions of the CLD. **d,** Denaturing urea–PAGE of WT and IF-del, *Dm*Cas7-11–crRNA incubated with labeled target RNA in the absence and presence of *Dm*TPR-CHAT at equimolar (1×) or double (2×) concentrations (n = 3). The red arrows indicate the cleavage products. **f**, Denaturing urea–PAGE of *Dm*Cas7-11–crRNA incubated with labeled target RNA in the absence and presence of WT, CTD-del and CTD-alone *Dm*TPR-CHAT (n = 3). **g**, Proposed model of *Dm*Cas7-11 regulation by *Dm*TPR-CHAT.

and Extended Data Fig. 10a–c,e,f). Of note, we observed, in sequence alignments of Cas7-11 from different species, that the insertion finger is conserved in multiple species, including *Dm*Cas7-11 and *Di*Cas7-11, but is absent in *Sb*Cas7-11 (Extended Data Fig. 10d and Supplementary Fig. 4). As the insertion finger is crucial for interacting with the CLD and stabilizing the closed conformation, we tested the effect of deletion of the insertion finger of *Dm*Cas7-11 (IF-del) on *Dm*TPR-CHAT-mediated inhibition. Strikingly, we found that *Dm*TPR-CHAT did not inhibit IF-del when compared with wild-type (WT) DmCas7-11 (Fig. 2d). We also show that the inhibitory role of *Dm*TPR-CHAT is mediated by its NTD. Deletion of the CTD (CTD-del) does not affect inhibition in contrast to the CTD alone, which lacks the NTD and does not inhibit *Dm*Cas7-11 (Fig. 2f and Extended Data Fig. 10c,e).

Based on our structural and biochemical analyses, we propose a model of Cas7-11 regulation by TPR-CHAT via stabilizing interactions between the insertion finger and CLD of *Dm*Cas7-11 and the NTD of *Dm*TPR-CHAT. The CLD then transiently blocks target RNA binding and reduces the Cas7-11 nuclease activity (Fig. 2g). The proposed mechanism is most likely to be conserved in other species, where the insertion finger and NTD are present in the Craspase complex, opening a path toward further engineering of the system to regulate Cas7-11 activity in future applications.

## Online content

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

## Methods

### Plasmid constructs design and cloning

DmCas7-11 used in the present study was coexpressed together with crRNA in *Escherichia coli* BL21(DE3) from a single plasmid containing the kanamycin resistance gene. CrRNA is processed to shorter crRNA molecules on coexpression with Ca7-11. To construct the plasmid for coexpression, a coding sequence of full-length DmCas7-11 codon optimized for *E. coli* was synthesized and cloned (GenScript Biotech) into MCS-1 of a pRSFDuet-1 plasmid, using BamHI and SalI restriction enzyme sites. DmCas7-11 was inserted in-frame with an N-terminal 6× Histidine (6×His)-tag or N-terminal FLAG tag.

CrRNA from *D. magnum*, composed of an array of five 72-bp repeats of the native sequence, each containing the first native spacer sequence and a sixth shorter repeat without spacer, was cloned into MCS-2 using BglII and XhoI restriction enzyme sites. Both DmCas7-11 and crRNA were inserted downstream of a LacI-repressed T7 promoter with the coding sequence DmCas7-11, terminating via a stop codon, and crRNA terminating via the T7 terminator.

*D. magnum* TPR-CHAT used in the present study was expressed in *E. coli* BL21(DE3) from a plasmid containing the ampicillin resistance gene. The coding sequence of full-length *D. magnum* TPR-CHAT with N-terminal Twin-strep-tag was synthesized and cloned into pETDuet-1 using the NcoI and XhoI restriction enzyme sites (GenScript Biotech). In this way, the gene was inserted downstream of a LacI-repressed T7 promoter to facilitate inducible expression using isopropyl β-ᴅ-1-thiogalactopyranoside (IPTG).

### Protein expression and purification

The *D. magnum* Craspase complex, composed of DmCas7-11–crRNA and DmTPR-CHAT, DmCas7-11–crRNA alone and DmTPR-CHAT alone, was expressed and purified from *E. coli* BL21(DE3). For expression of the Craspase complex, chemically competent *E. coli* BL21(DE3) was transformed with both the pRSFDuet-1 plasmid containing the kanamycin resistance gene for expression of DmCas7-11 together with crRNA, and the pETDuet-1 plasmid containing the ampicillin resistance gene for the expression of DmTPR-CHAT, and grown overnight at 37 °C on Luria broth (LB) agar plates containing both selection antibiotics (50 µg ml$^{-1}$ of kanamycin and 100 µg ml$^{-1}$ of carbenicillin). The colonies obtained were streaked from the plate and transferred to 50 ml of 2xYT medium containing the selection antibiotics and grown overnight at 37 °C with shaking. Then, 40 ml of the overnight culture was used to inoculate 4 l of 2xYT medium containing the selection antibiotics. The cultures were grown at 37 °C with shaking at 190 r.p.m. until they reached an absorbance at 600 nm of 0.5–0.7, then incubated on ice for 1 h, before inducing protein expression with 0.5 mM IPTG for 18 h at 20 °C. The overnight cultures were harvested by centrifugation at 3,000g for 30 min at 4 °C. The resulting supernatant was discarded and the pellet was resuspended in 100 ml of cold lysis buffer (25 mM Hepes-NaOH, pH 7.5, 200 mM NaCl, 10% glycerol, 25 mM imidazole and 1 mM 2-mercaptoethanol) supplemented with two tablets of cOmplete EDTA-free Protease Inhibitor Cocktail (Roche) and 6 µg ml$^{-1}$ of RNAseA (Thermo Fisher Scientific) before lysis by sonication. The lysate was clarified by centrifugation for 30 min at 70,560g and 4 °C in an Optima XPN Ultracentrifuge (Beckman Coulter) using a Ti-45 rotor. The supernatant, which contained soluble 6xHis-tagged DmCas7-11 in complex with crRNA and Twin-strep-tagged DmTPR-CHAT, was loaded on to 3 ml of HisPur Ni-NTA Resin (Thermo Fisher Scientific) pre-equilibrated with wash buffer (25 mM Hepes-NaOH, pH 7.5, 200 mM NaCl, 10% glycerol, 35 mM imidazole and 1 mM 2-mercaptoethanol) in an XK16/20 column (Cytiva Life Sciences). After loading, the column was washed with 25 ml of wash buffer and eluted with 10 ml of elution buffer (25 mM Hepes-NaOH, pH 7.5, 200 mM NaCl, 10% glycerol, 250 mM imidazole and 1 mM 2-mercaptoethanol). Pooled elution fractions were concentrated to ~500 ul in 100K Amicon Ultra-15 concentrators (Millipore) and further purified by gel filtration chromatography on a 10/300

GL Superose 6 gel filtration column (Cytiva Life Sciences) in gel filtration buffer (25 mM Hepes-NaOH, pH7.5, 150 mM NaCl and 1 mM dithiothreitol (DTT)). Peak fractions (as determined by the chromatograms with ultraviolet light of 280 nm) generated from the Unicorn software (v.7.1) containing complete Craspase complexes of subunits DmCas7-11–crRNA and DmTPR-CHAT (as determined by sodium dodecylsulfate (SDS)–polyacrylamide gel electrophoresis (PAGE) analysis), were pooled and concentrated to an absorbance at 280 nm of 0.8 to prepare cryo-EM grids. Craspase complex for use in biochemical experiments was purified in gel filtration buffer supplemented with 10% glycerol with peak fractions pooled, concentrated, flash-frozen and stored at −80 °C.

For the expression of full-length insertion finger and IF-del mutant of DmCas7-11–crRNA alone, chemically competent *E. coli* BL21(DE3) was transformed with a pRSFDuet-1 plasmid containing DmCas7-11 together with crRNA and kanamycin resistance gene, and grown overnight at 37 °C on LB agar plates supplemented with antibiotic (50 µg ml$^{-1}$ of kanamycin). Downstream expression and purification were done using the same procedure described for the Craspase complex with the exception that RNAse A was omitted from the lysis buffer. However, for biochemical experiments, an N-terminal FLAG tag version of DmCas7-11 was used. Therefore purification was performed using anti-FLAG.

M2 affinity gel (Millipore): the beads were washed with 100 ml of wash buffer (25 mM Hepes-NaOH, pH 7.5, 200 mM NaCl, 10% glycerol and 1 mM 2-mercaptoethanol) and eluted with 4 ml of elution buffer (25 mM Hepes-NaOH, pH 7.5, 200 mM NaCl, 10% glycerol, 120 µg ml$^{-1}$ of 3× FLAG peptide and 1 mM 2-mercaptoethanol), followed by buffer exchange and concentration on a 100K Amicon Ultra-15 concentrators (Millipore).

For the expression of full-length, CTD-del and CTD of DmTPR-CHAT alone, chemically competent *E. coli* BL21(DE3) was transformed with a pETDuet-1 plasmid containing DmTPR-CHAT and the ampicillin resistance gene and grown overnight at 37 °C on LB agar plates supplemented with antibiotic (50 µg ml$^{-1}$ of kanamycin). The remaining procedure of protein expression was carried out as already described for the Craspase complex. For purification, the harvested cell pellet was resuspended in 100 ml of cold lysis buffer (25 mM Hepes-NaOH, pH 7.5, 200 mM NaCl, 10% glycerol and 1 mM 2-mercaptoethanol) supplemented with two tablets of cOmplete EDTA-free Protease Inhibitor Cocktail before lysis by sonication. The lysate was clarified by centrifugation for 30 min at 70,560g and 4 °C in an Optima XPN Ultracentrifuge using a Ti-45 rotor. The supernatant, which contained soluble Twin-strep-tagged DmTPR-CHAT, was loaded on to 2 ml of Streptactin Sepharose High-Performance resin (Cytiva Life Sciences) pre-equilibrated with wash buffer (25 mM Hepes-NaOH, pH7.5, 200 mM NaCl, 10% glycerol and 1 mM 2-mercaptoethanol) on an XK16/20 column. After loading, the column was washed with 20 ml of wash buffer, 10 ml of high-salt wash buffer (25 mM Hepes-NaOH, pH 7.5, 800 mM KCl, 10% glycerol and 1 mM 2-mercaptoethanol) and another 10 ml of wash buffer before elution with 10 ml of elution buffer (25 mM Hepes-NaOH, pH 7.5, 200 mM NaCl, 10% glycerol, 5 mM desthiobiotin and 1 mM 2-mercaptoethanol). Pooled elution fractions were concentrated to ~500 µl in 30K Amicon Ultra-15 concentrators and further purified by gel filtration chromatography on a 10/300 GL Superose 6 gel filtration column in gel filtration buffer (25 mM Hepes-NaOH pH7.5, 150 mM NaCl, 10% glycerol and 1 mM DTT). Peak fractions containing DmTPR-CHAT, as determined by SDS–PAGE analysis, were pooled, concentrated, flash-frozen and stored at −80 °C.

### SDS–PAGE analysis

To assess the purity of purified proteins and the presence of individual subunits in the complex, an SDS–PAGE analysis was performed; 15 µl of protein sample was supplemented with 5 µl of 4× NuPAGE LDS Sample Buffer (Thermo Fisher Scientific). Samples were incubated at 95 °C for 10 min before loading on a 4–12% NuPAGE Bis–Tris Precast Gel (Thermo

Fisher Scientific). PageRuler Prestained Protein Ladder (10–180 kDa) was also loaded on the gel to run as a size marker. Gels were run in 1× NuPAGE MES SDS running buffer (Thermo Fisher Scientific) at 200 V for 30 min, washed briefly in MilliQ water and stained for 2 h with QuickBlue Protein Stain (LuBioScience GmbH) with shaking. Gels were washed in MilliQ water before imaging on an iBright FL1500 Imaging System (Thermo Fisher Scientific). Gel images were processed and prepared on ImageJ (v.1.53k).

### In vitro target RNA cleavage assay

To investigate the target RNA cleavage activity of *Dm*Cas7-11–crRNA in the presence and absence of *Dm*TPR-CHAT, in vitro RNA cleavage assays were performed using 48 bp of synthesized 5′-Cy5-labeled single-stranded (ss)RNA (Microsynth) as a substrate. RNA cleavage reactions were performed with 50 nM labeled ssRNA incubated with 50–200 nM purified *Dm*Cas7-11–crRNA in nuclease assay buffer (32.5 mM Hepes-NaOH pH 7.5, 100 mM NaCl and 2.5 mM DTT) supplemented with 1 U μl⁻¹ of RNAse inhibitor (New England Biolabs). Then, 200 nM purified *Dm*TPR-CHAT was included in the reaction where indicated. Reactions were incubated at 37 °C for 1 h and quenched by adding urea and proteinase K (Thermo Fisher Scientific) at final concentrations of 0.1 M and 1 μg μl⁻¹, respectively, and incubated at 50 °C for 15 min. The 2× loading dye (1× tris-borate–EDTA (TBE), 12% Ficoll, 7 M urea) was added to the reaction to a final 1× concentration, followed by heating at 95 °C for 5 min before loading on a 15% Novex PAGE TBE–urea gel (Thermo Fisher Scientific). The gel was run at 200 V for 45 min and imaged on an iBright FL1500 Imaging System. To investigate the inhibition of target RNA cleavage activity of *Dm*Cas7-11–crRNA, equimolar or twofold excess purified *Dm*TPR-CHAT constructs were incubated with *Dm*Cas7-11–crRNA for 1 h on ice before the addition of labeled RNA. The reactions were incubated at 37 °C for 15 min or 1 h when indicated and further processed as already described. Gel images were processed and prepared on ImageJ (v.1.53k).

### In vitro target RNA-binding assay

To investigate the target RNA binding of *Dm*Cas7-11–crRNA in the presence and absence of *Dm*TPR-CHAT, electrophoresis mobility shift assays were performed with 50 nM 5′-Cy5-labeled ssRNA incubated with 1 μM purified *Dm*Cas7-11–crRNA in binding buffer (32.5 mM Hepes-NaOH pH 7.5, 100 mM NaCl and 2.5 mM DTT) supplemented with 1 U μl⁻¹ of RNAse inhibitor. The reaction was incubated for 30 min on ice before the addition of 2× loading dye (15% sucrose) to a final 1× concentration. The sample was then directly loaded on Novex TBE gels, 4–12% (Thermo Fisher Scientific). The gel was run at 100 V for 90 min and imaged on an iBright FL1500 Imaging System. To investigate the effect on target RNA binding of *Dm*Cas7-11–crRNA, equimolar 200 nM purified *Dm*TPR-CHAT constructs were incubated with *Dm*Cas7-11–crRNA for 1 h on ice before the addition of labeled RNA and the remaining steps of the assay were carried out. Gel images were processed and prepared on ImageJ (v.1.53k).

### Negative stain analysis of *D. magnum* Craspase complex

The purified *D. magnum* Craspase complex and *Dm*Cas7-11–crRNA were first analyzed by negative staining to check for sample quality before cryo-EM sample preparation. Of the diluted sample at 80–100 nM, 3.5 μl was applied to glow-discharged 400-mesh copper grids coated with a continuous carbon film (Electron Microscopy Sciences). The grids were stained by incubating with 2% (w:v) uranyl acetate for a total of 30 s, then blotted and air-dried. Images were collected on a Philips CM100 Biotwin transmission electron microscope, operating at 80 kV, with a TVIPS F416 CMOS camera (4,000 × 4,000) at a physical pixel size of 0.6 nm at the sample level, with a total electron dose of approximately 4 e⁻ Å⁻² over a total exposure time of 400 ms.

### Cryo-EM sample preparation and data collection

Cryo-EM grids were prepared by applying 3.5 μl of concentrated sample on to 400-mesh R1.2/1.3 UltrAuFoil grids (Quantifoil Micro Tools GmbH), which had been rendered hydrophilic by glow discharging at 15 mA for 60 s with a PELCO easiGlow device (Ted Pella, Inc.). The sample was adsorbed for 30 s on the grids, followed by blotting and plunge freezing into liquid ethane using a Vitrobot Mark IV plunge freezer (Thermo Fisher Scientific). Cryo-EM data were collected using the automated data acquisition software EPU (Thermo Fisher Scientific) on a Titan Krios G4 transmission electron microscope (Thermo Fisher Scientific), operating at 300 kV and equipped with a cold field emission gun electron source and a Falcon4 direct detection camera. Images were recorded in counting mode at a nominal magnification of ×96,000, corresponding to a physical pixel size of 0.83 Å at the sample level. Datasets were collected at a defocus range of 0.8–2.5 μm with a total electron dose of 60 e⁻/Å⁻². Image data were saved as electron event recordings.

### Cryo-EM image processing, model building and refinement

The cryo-EM image processing was performed using cryoSPARC v.3.3 (ref. [16]).

The patch-based motion correction (cryoSPARC implementation) was used for aligning the EM video stacks, as well as applying dose-dependent resolution weighting to recorded videos. Contrast transfer function estimation was performed using the patch-based option as well. For the data of the Craspase (*Dm*Cas7-11–TPR-CHAT) complex, 2,000 particles were manually picked and used for one round of 2D classification for template creation. Template-based automated particle picking was then used on the recorded image data, which resulted in a set of 1,719,283 particles. Two rounds of 2D classification were performed for the initial step of particle cleaning. Multiple rounds of 2D classifications were performed for more specific 2D rebalancing, by deselecting the most abundant views manually, resulting in a particle set of 887,670 particles. Initial reconstruction and heterorefinement yielded multiple three-dimensional (3D) classes. Two classes accounting for 29% of the total particles were removed because they showed particles of a highly preferred orientation. The other 3D classes were grouped into two major classes for further processing, splitting the particles into one class with clear density for the entire TPR-CHAT (*Dm*Cas7-11 and *Dm*TPR-CHAT_full) and another class with less density in the TPR-CHAT-binding area (*Dm*Cas7-11 and *Dm*TPR-CHAT_NTD). The signs of preferred orientation of the particles in the remaining *Dm*Cas7-11–TPR-CHAT_full class were still noticeable after 3D refinement, so an additional step of 2D rebalancing was performed to further reduce this anisotropic effect. After this selection, 214,592 particles remained, which were subjected to 3D classification, resulting in 6 classes. The best 3D class consisting of 65,755 particles was refined and yielded a cryo-EM map at 3.20-Å overall resolution in C1 symmetry.

For the classes *Dm*Cas7-11 and TPR-CHAT_NTD, a total of 176,623 particles were classified by an additional heterorefinement and then subjected to 3D classification, resulting in another three classes. The best 3D class containing 53,969 particles produced a cryo-EM map in C1 symmetry at an overall resolution of 3.03 Å. Subsequently, local refinement was performed with an insertion region-specific mask volume. The resolution for the locally refined cryo-EM map was estimated at 3.43 Å in C1 symmetry. All resolution measures used the Fourier shell correlation (FSC) criterion of a 0.143 cutoff[17].

Atomic models for both *Dm*Cas7-11–crRNA and *Dm*TPR-CHAT_full and *Dm*Cas7-11–crRNA and *Dm*TPR-CHAT_NTD structures were mostly newly built manually in Coot 0.9.4 (ref. [18]) For the regions of low density or low quality, an AlphaFold2 (ColabFold implementation) prediction was used as a guide for amino acid assignment and backbone tracing[19]. Real-space refinement for all built models was performed using Phenix v.1.19.2-4158 by applying a general restraints set-up[20]. Structural alignments and superpositions were performed with PyMOL (PyMOL

Molecular Graphics System v.2.0, Schrödinger, LLC). Figures were created using PyMOL, UCSF Chimera, UCSF ChimeraX[21] and Adobe Illustrator (https://adobe.com/products/illustrator).

### Multiple sequence alignments

Multiple protein sequence alignments were performed using Clustal Omega (http://www.ebi.ac.uk/Tools/msa/clustalo) and visualized in JALVIEW[22]. Phylogenetic trees were prepared using the neighbor-joining algorithm on Geneious prime (v.2022.0.2, Biomatters).

### Reporting summary

Further information on research design is available in the Nature Portfolio Reporting Summary linked to this article.

### Data availability

The reconstructed maps are available from the Electron Microscopy Data Bank (EMDB) database under accession nos. EMDB-14847 and EMDB-14848. The atomic models are available in the Protein Databank (PDB) database, accession nos. 7ZOL and 7ZOQ. The raw video data of this work are available in the Electron Microscopy Public Image Archive database, accession no. EMPIAR-11268 (https://doi.org/10.6019/EMPIAR-11268). Source data are provided with this paper.

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

### Acknowledgements

This work was supported by the Swiss National Science Foundation (grant no. CRSII5_177195) and the NCCR TransCure (grant no. 185544) awarded to H.S. We thank the Dubochet Center for Imaging from the EPFL, University of Lausanne and University of Geneva for their support.

### Author contributions

D.N. and B.E. conceived and designed the study. B.E., D.N. and D.T. performed experiments. B.B., S.N. and A.M collected cryo-EM data. D.N. and B.E. performed formal analysis. B.E. and D.N. prepared, reviewed and edited the manuscript. H.S. acquired funding and reviewed the manuscript. H.S. and D.N. supervised the study. All authors gave their input in the preparation of the final manuscript.

### Competing interests

The authors declare no competing interests.

### Additional information

**Extended data** is available for this paper at https://doi.org/10.1038/s41594-022-00894-5.

**Correspondence and requests for materials** should be addressed to Henning Stahlberg or Dongchun Ni.

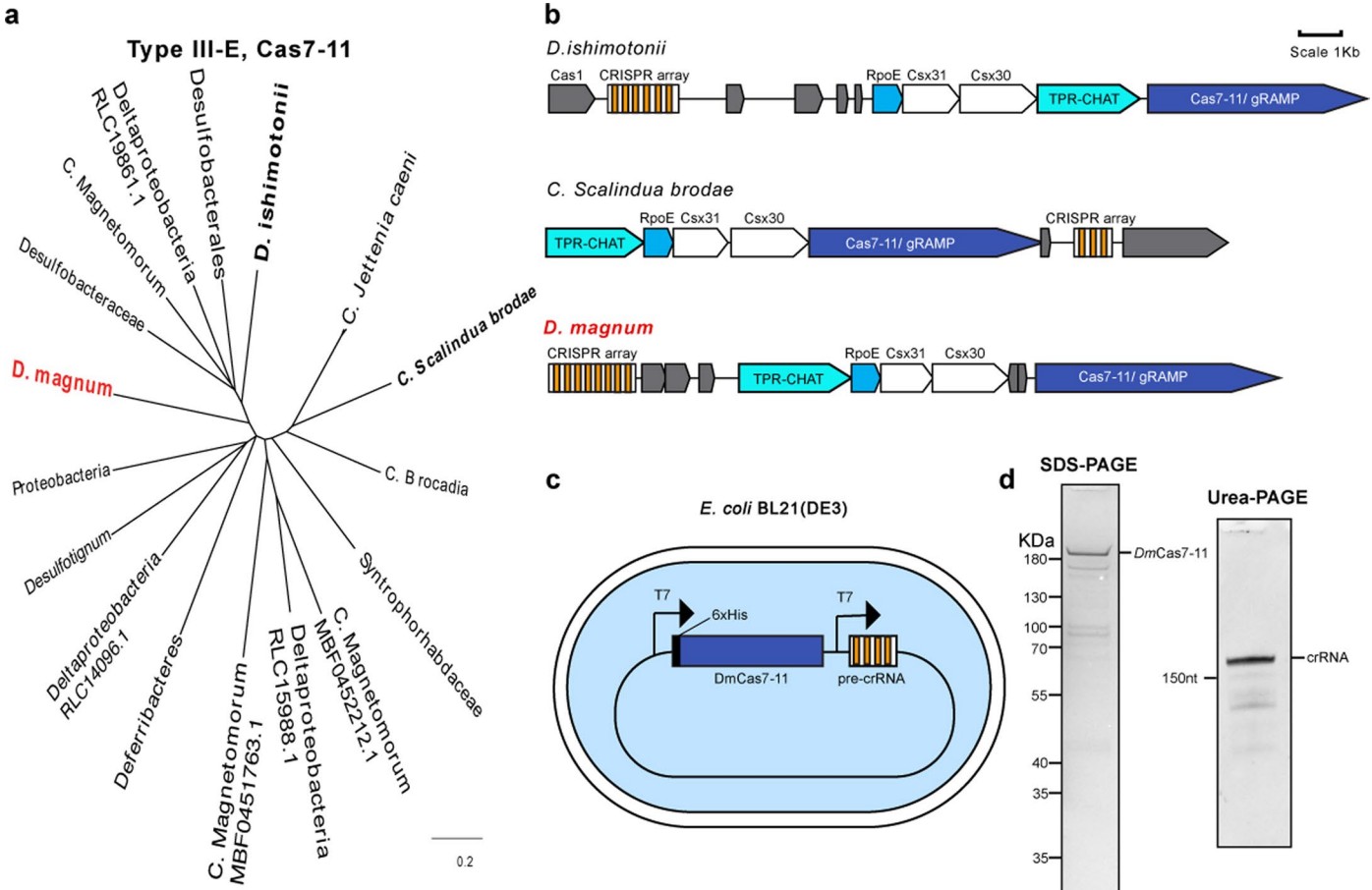

**Extended Data Fig. 1 | Description and Purification of *D. magnum* Cas7-11-crRNA. a**) Phylogenetic tree of type III-E CRISPR effectors. **b**) Schematic representation of the Type−IIIE loci from *D. magnum*, *D. ishimotonii* and *S. brodae*. **c**) Schematic representation of *Dm*Cas7-11-crRNA recombinant expression in *E. coli* BL21 (DE3). **d**) The left panel shows an SDS-PAGE of *Dm*Cas7-11, and the right panel shows a Urea-PAGE of crRNA from purified *Dm*Cas7-11-crRNA (n = 6).

**a**

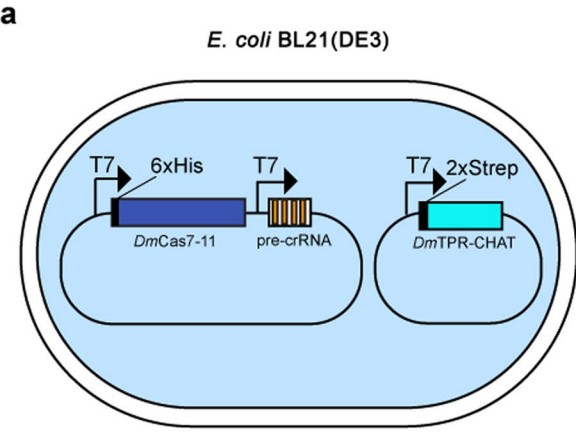

**b**

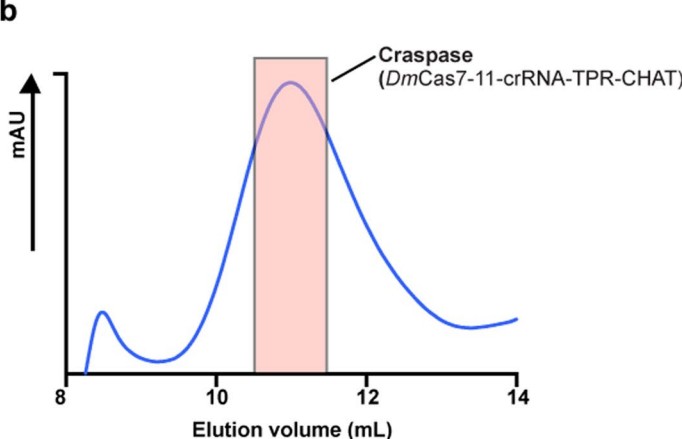

**c**

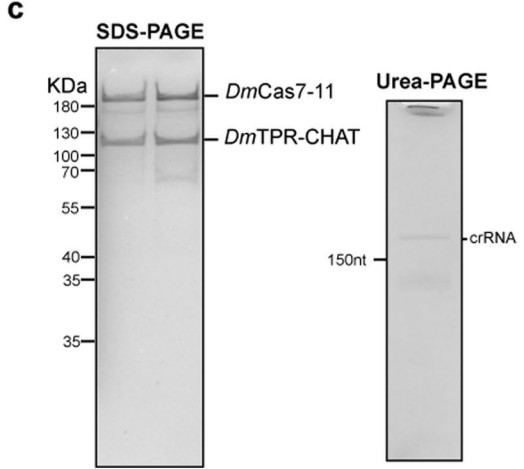

**d**

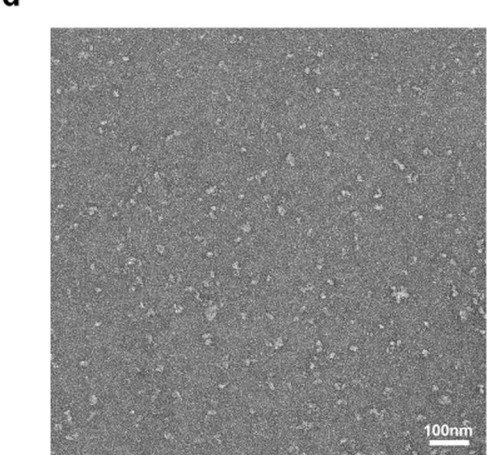

**Extended Data Fig. 2 | Purification of the *D. magnum* Craspase complex (*Dm*Cas7-11-crRNA and TPR-CHAT). a**) Schematic representation of *Dm*Cas7-11-crRNA and TPR-CHAT recombinant expression in *E. coli* BL21 (DE3). **b**) Size exclusion chromatogram from the purification of the *D. magnum* Craspase complex. Peak fractions pooled for structural studies are indicated by a pink box. **c**) The left panel shows an SDS-PAGE of *Dm*Cas7-11 and *Dm*TPR-CHAT, and the right panel shows a Urea-PAGE of crRNA from the purified Craspase complex (n = 4). **d**) EM micrograph of a negatively stained preparation of purified Craspase complexes (n = 22).

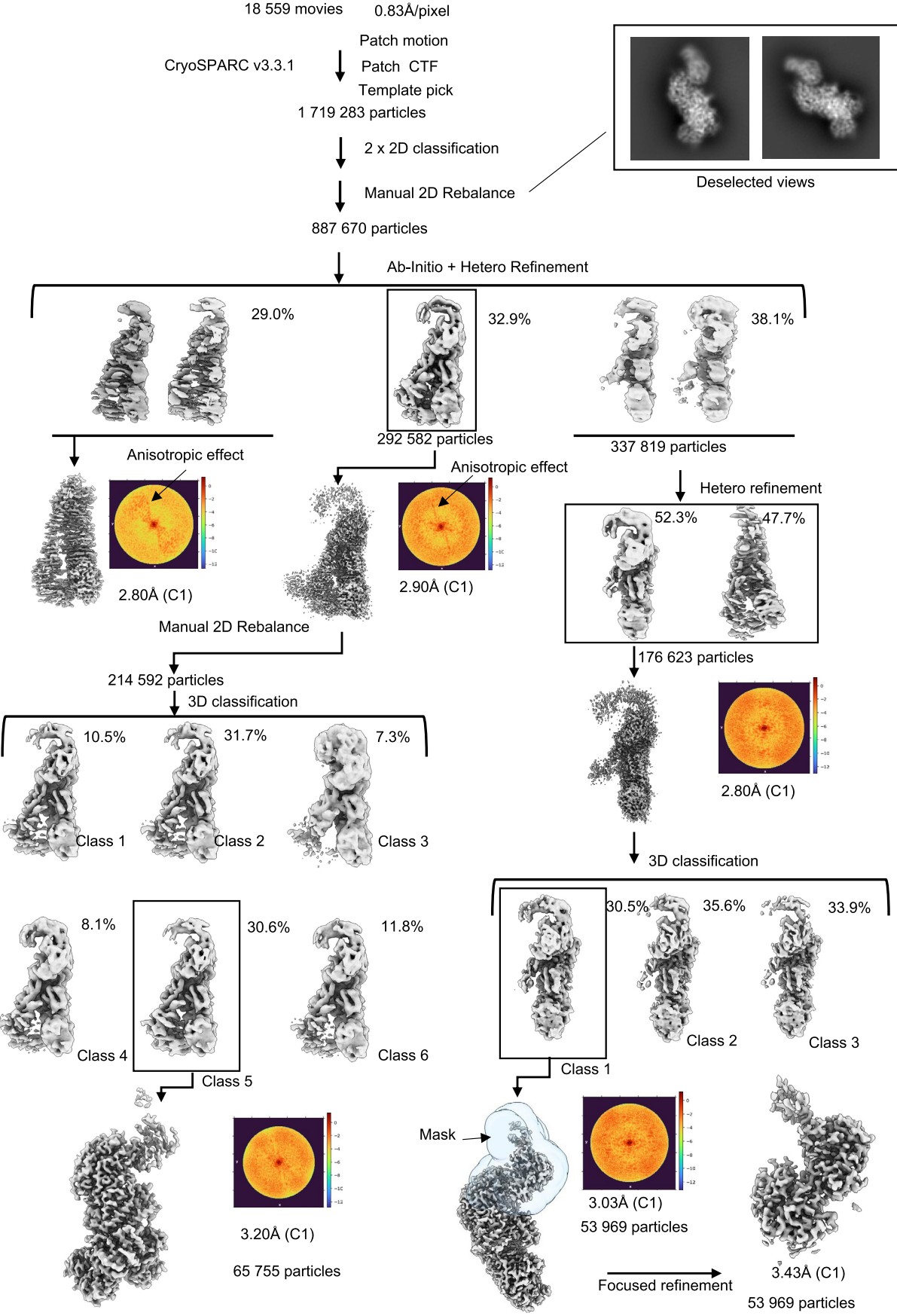

**Extended Data Fig. 3 | Cryo-EM data processing workflow for *D. magnum* Craspase complex.** Processing workflow for *Dm*Cas7-11-crRNA-*Dm*TPR-CHAT_full and *Dm*Cas7-11-crRNA-*Dm*TPR-CHAT_NTD showing 2D class averages of deselected views, 3D classification and refinement steps. Focused refinement with a mask for the insertion domains is also shown.

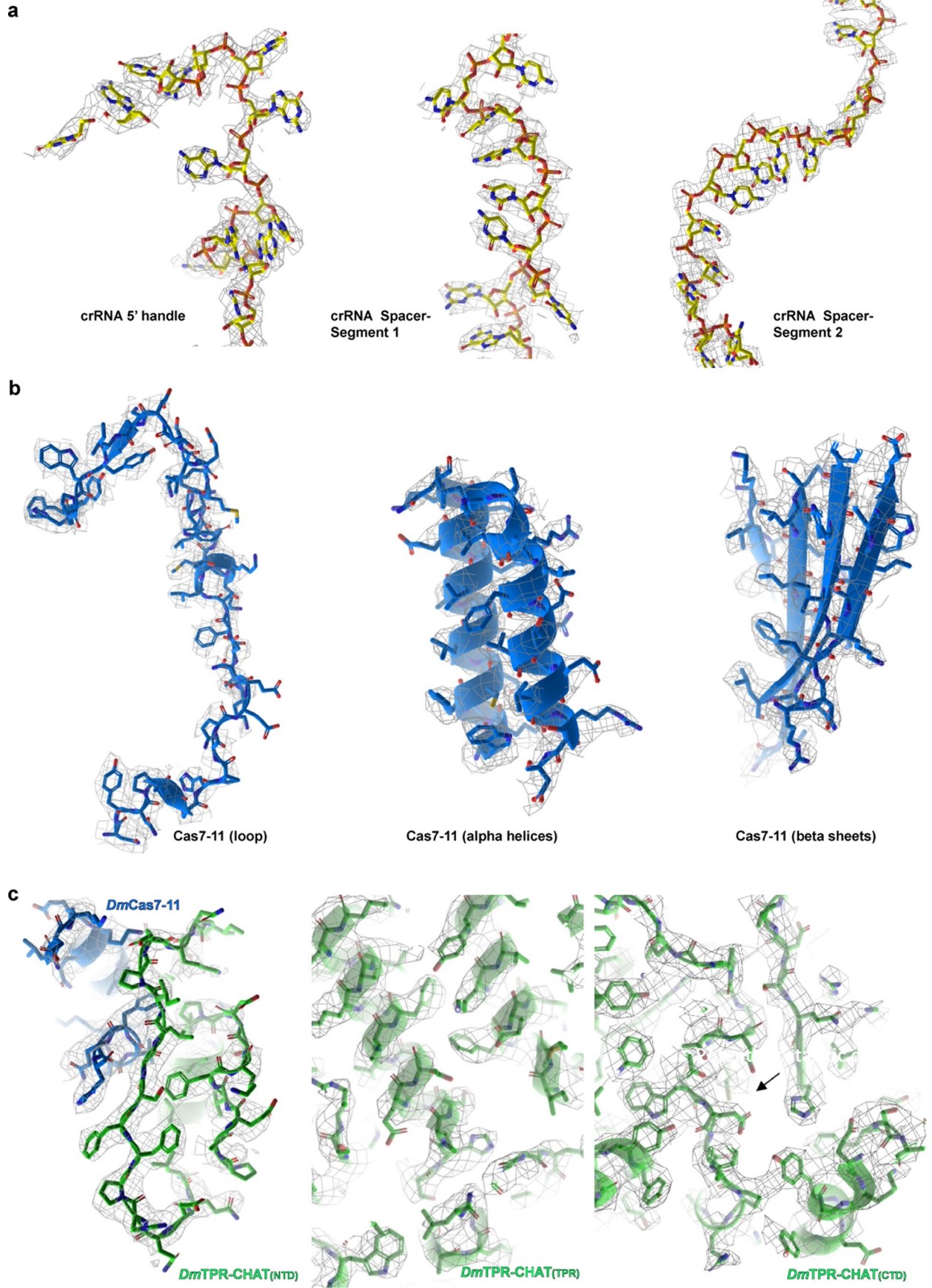

**Extended Data Fig. 4 | Zoom-in view of the cryo-EM maps of structural elements of crRNA (a), *Dm*Cas7-11 (b), and *Dm*TPR-CHAT (c). a)** Cryo-EM map densities and atom displays for the crRNA 5' handle, crRNA spacer-Segment 1 and crRNA-Segment 2. **b)** Cryo-EM map densities and atom displays for a loop, alpha helices and beta sheets of *Dm*Cas7-11. **c)** Cryo-EM map densities and atom displays for regions of the NTD, TPR and CTD of *Dm*TPR-CHAT.

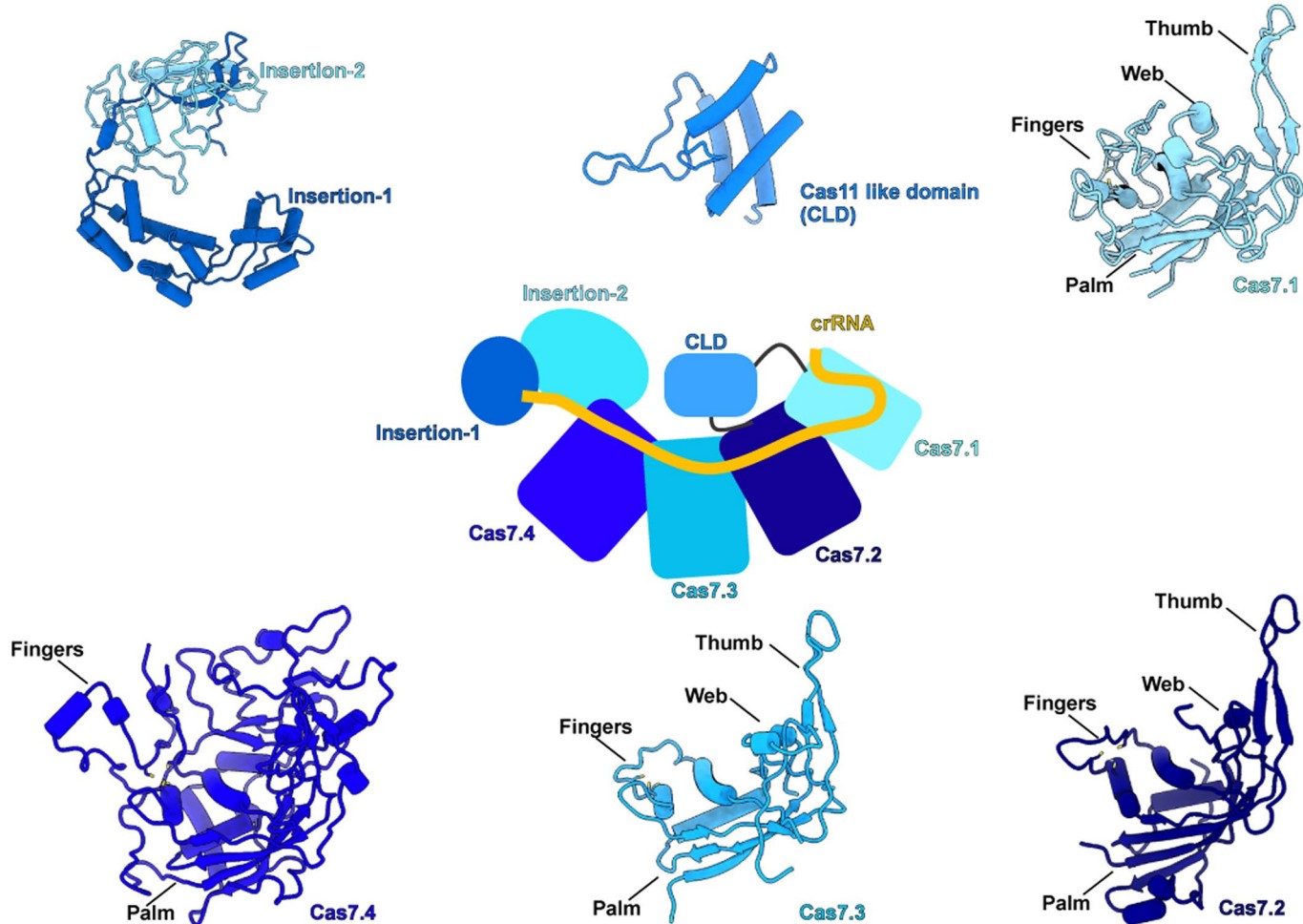

**Extended Data Fig. 5 | Domain annotation of the Cas7 repeats.** Schematic representation of the domain organization of *Dm*Cas7-11-crRNA (center) surrounded by cartoon representations of the Insertion-1, Insertion-2, CLD, Cas7.1, Cas7.2, Cas7.3 and Cas7.4 domains. The thumb, web, fingers and palm of the Cas7 domains are indicated.

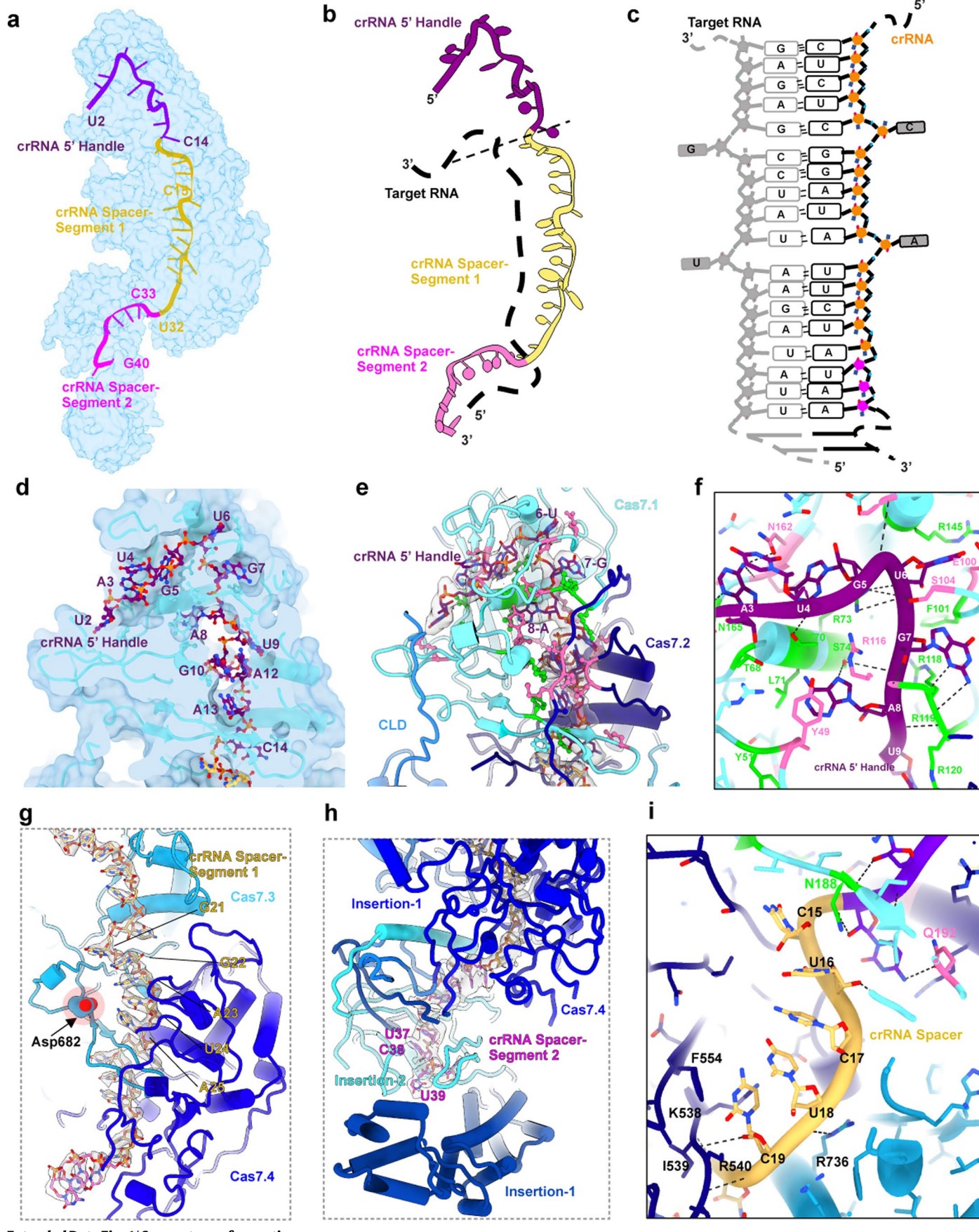

**Extended Data Fig. 6 | See next page for caption.**

**Extended Data Fig. 6 | Structure of crRNA bound to Cas7-11. a)** Structure representation of crRNA bound to Cas7-11 indicating the positions of the 5′ handle and spacer RNA segments. **b, c)** Cartoon representation of target RNA bound to crRNA spacer. **d-f)** Detailed structural representation of the interaction between the 'hook-like' 5′ handle and Cas7.1 and Cas7.2 domains. **g-i)** Detailed structural representation of the interaction between the crRNA spacer and Cas7.1 and Cas7.2 domains. g,i) show the binding of crRNA spacer segment 1 to Cas7.3 and Cas7.4 domains. h) shows the binding of crRNA spacer segment 2 to the Cas7.4 Insertion-1 subdomain.

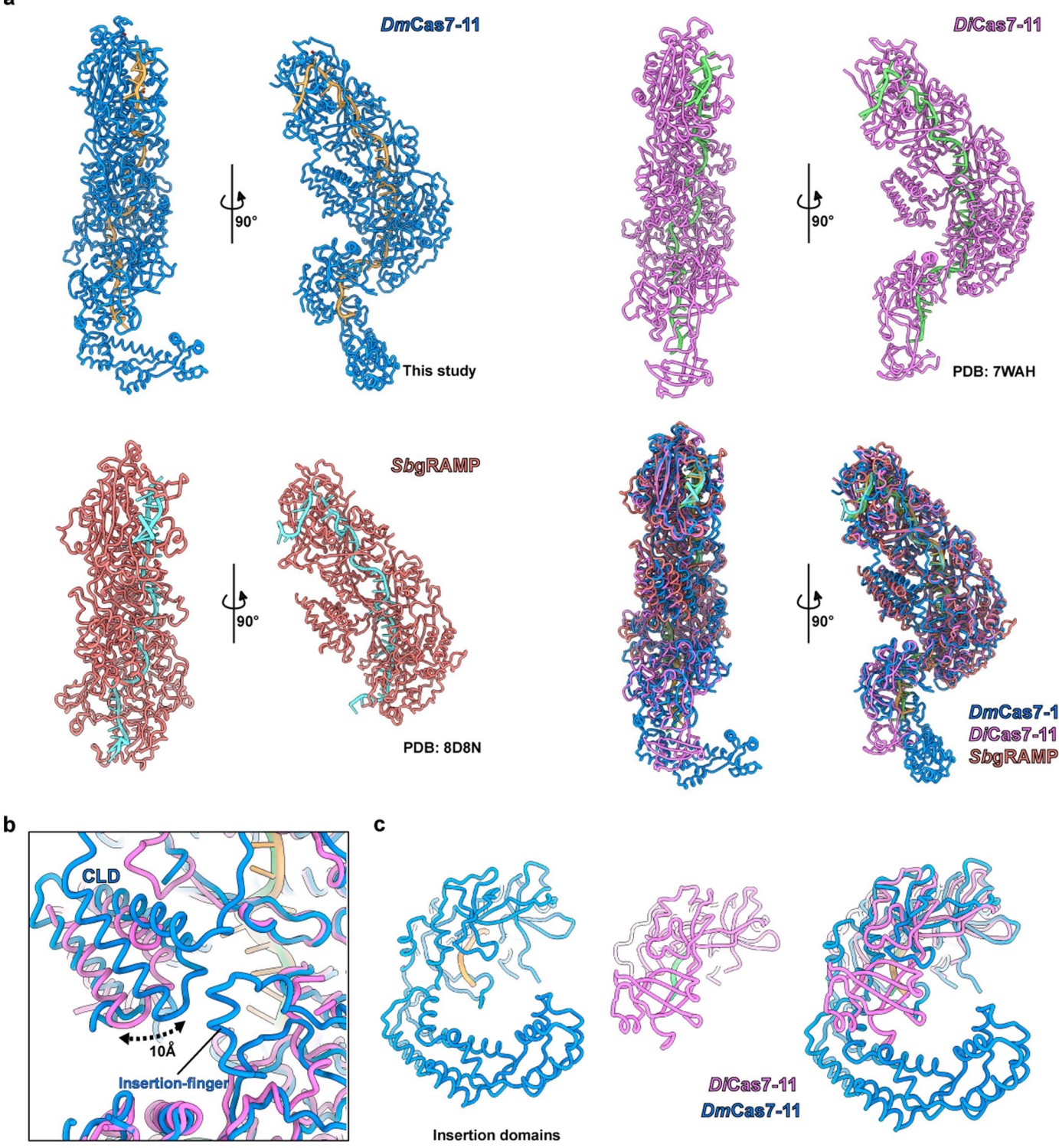

**Extended Data Fig. 7 | Structural alignment of Cas7-11 from different species.**
**a**) Cartoon representations of Cas7-11 from *D. magnum* (*Dm*Cas7-11- this study), *D. ishitimonii* (*Di*Cas7-11- PDB:7WAH) and *S. brodae* (*Sb*gRAMP- PDB:8D8N)[11,12]. **b**) Comparison of the position of the CLD between *Dm*Cas7-11 and *Di*Cas7-11 showing a 10 Å rotational movement. **c**) Structural comparison of the insertion domains between *Dm*Cas7-11 and *Di*Cas7-11.

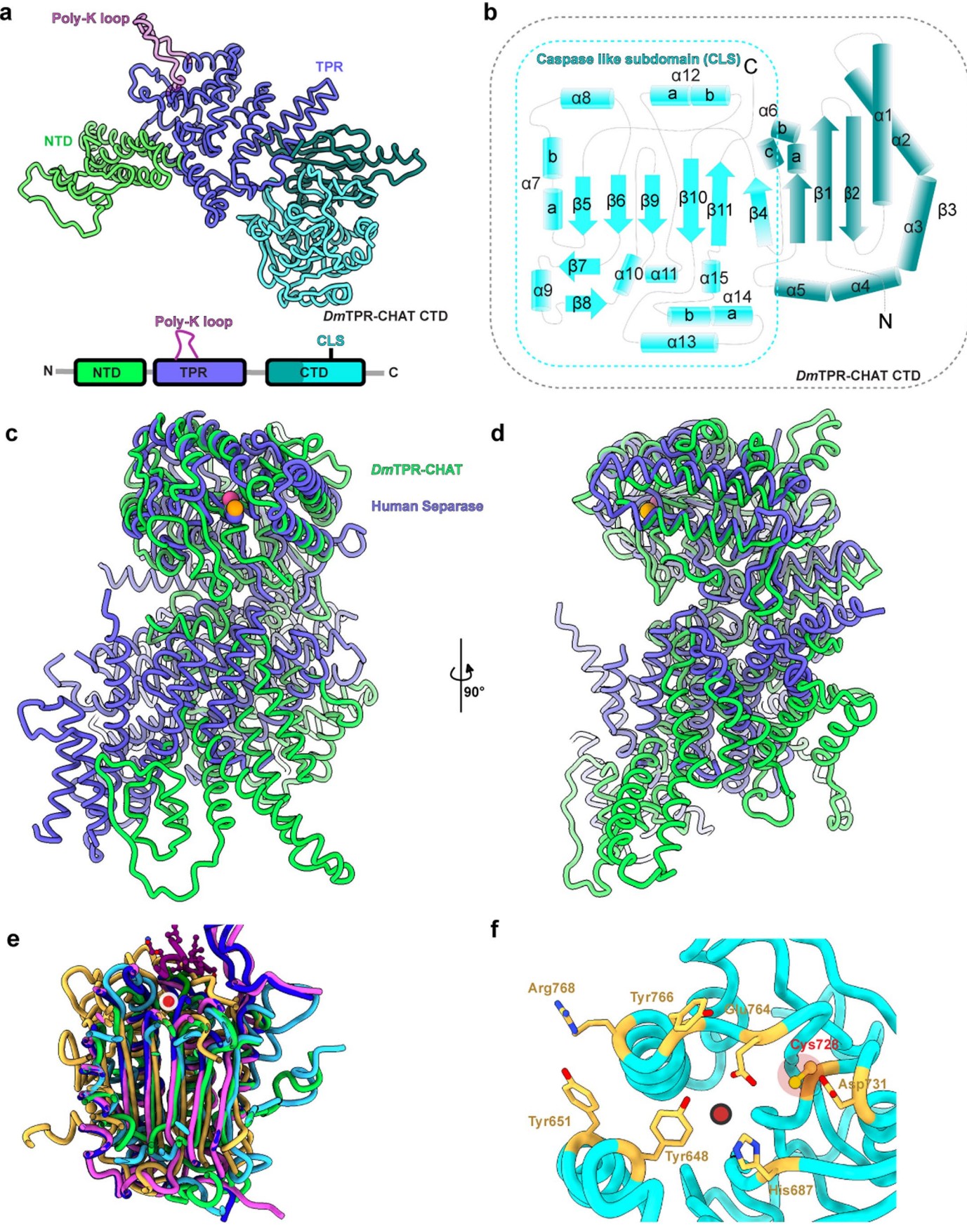

**Extended Data Fig. 8 | Structure of *Dm*TPR-CHAT. a**) Ribbon representation of *Dm*TPR-CHAT with domains distinctly coloured. **b**) Topology diagram of *Dm*TPR-CHAT CTD. **c**, **d**) Structure alignment between *Dm*TPR-CHAT and Human Separase (PDB ID: 7nj0). **e**) Structure alignment between the CLS (cyan), Seprase (green), Caspase7 (+/- substrate) (blue and purple), and PIGK (PDB IDs: 7nj0, 7w72, 1K88, and 1K86). The substrate binding site is shown as a red circle. **f**) Residues of the substrate–binding site in the CLS.

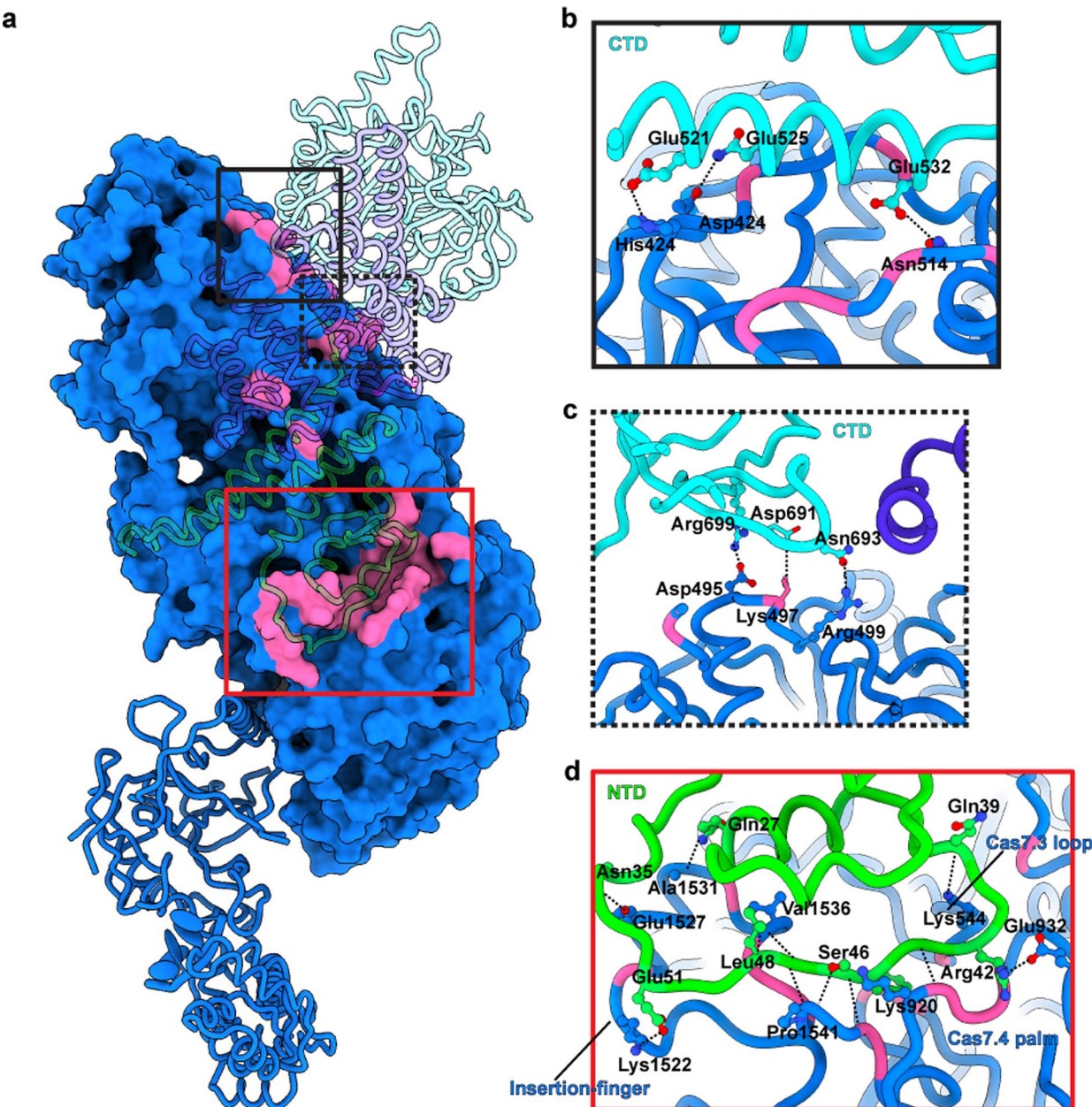

**Extended Data Fig. 9 | The interaction interfaces between *Dm*Cas7-11 and *Dm*TPR-CHAT.** a) Surface representation of the interaction interface between *Dm*Cas7-11 (blue surface and cartoon representation) and *Dm*TPR-CHAT (transparent cartoon representation) shown in pink. **b**–**d**) Detailed views of amino acid interactions at the interfaces corresponding to boxed areas in (a).

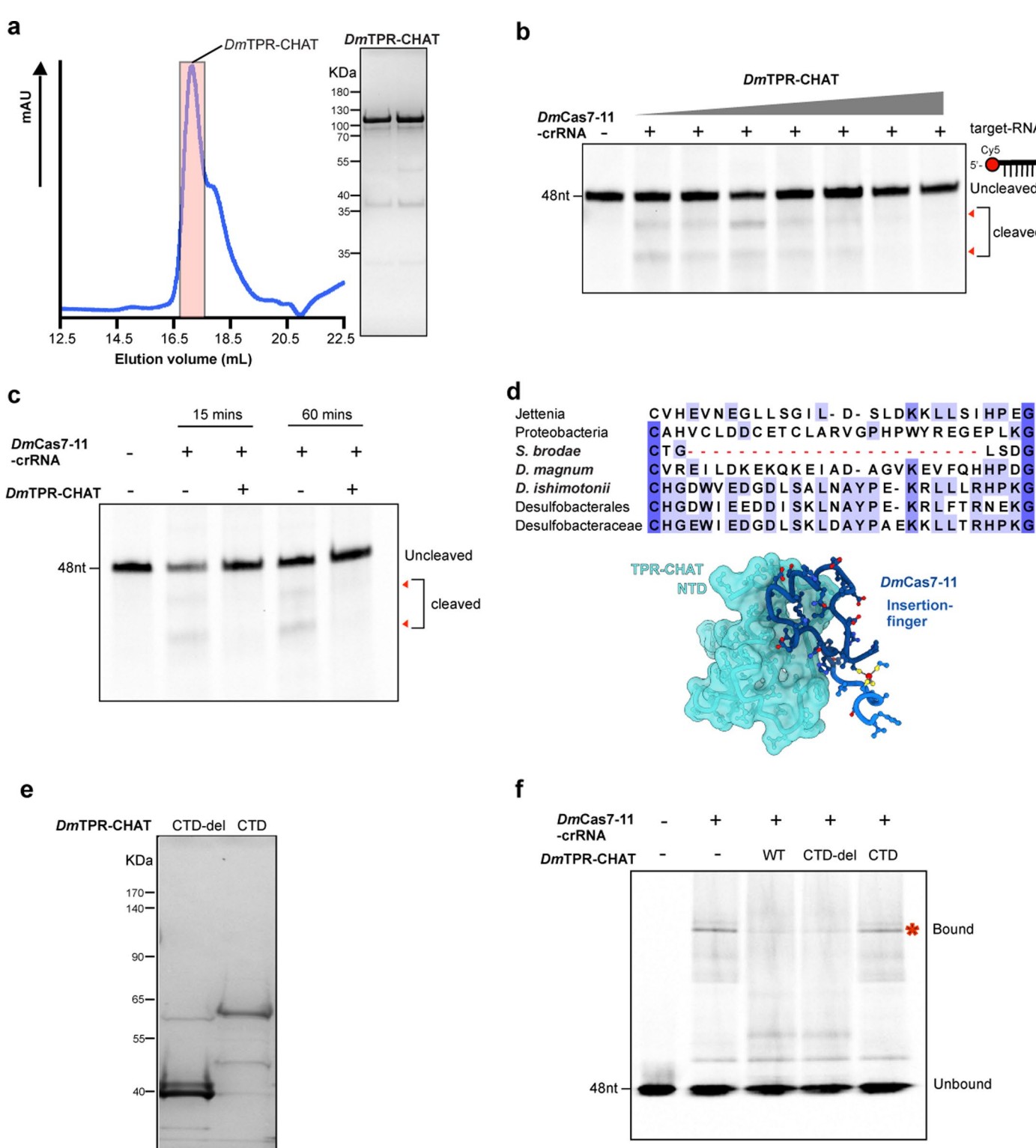

**Extended Data Fig. 10 | Inhibition of *Dm*Cas7-11 nuclease activity by *Dm*TPR-CHAT. a**) Chromatogram at 280 nm and SDS-PAGE gel for the purification of *Dm*TPR-CHAT (n = 3). **b**) Denaturing Urea-PAGE of *Dm*Cas7-11-crRNA incubated with 48bp-Cy5 labelled target and increasing concentrations of *Dm*TPR-CHAT. Red arrows indicate the cleavage products (n = 3). **c**) Denaturing Urea-PAGE of *Dm*Cas7-11-crRNA incubated with 48bp-Cy5 labelled in the presence and absence of *Dm*TPR-CHAT with 15 mins and 60 mins incubation times (n = 3). **d**) A sequence alignment of the Insertion-finger is shown in the top panel and represented as a ribbon bound to the NTD in the bottom panel. **e**) SDS-PAGE of CTD deletion (CTD-del) and CTD alone of *Dm*TPR-CHAT (n = 3). **f**) EMSA comparing the binding of *Dm*Cas7-11-crRNA to 48bp-Cy5 labelled target RNA in the absence and presence of full-length, CTD deletion and CTD alone *Dm*TPR-CHAT (n = 2).

# Reporting Summary

## Statistics

For all statistical analyses, confirm that the following items are present in the figure legend, table legend, main text, or Methods section.

| n/a | Confirmed | |
|---|---|---|
| ☐ | ☒ | The exact sample size (*n*) for each experimental group/condition, given as a discrete number and unit of measurement |
| ☐ | ☒ | A statement on whether measurements were taken from distinct samples or whether the same sample was measured repeatedly |
| ☒ | ☐ | The statistical test(s) used AND whether they are one- or two-sided <br> *Only common tests should be described solely by name; describe more complex techniques in the Methods section.* |
| ☒ | ☐ | A description of all covariates tested |
| ☒ | ☐ | A description of any assumptions or corrections, such as tests of normality and adjustment for multiple comparisons |
| ☒ | ☐ | A full description of the statistical parameters including central tendency (e.g. means) or other basic estimates (e.g. regression coefficient) AND variation (e.g. standard deviation) or associated estimates of uncertainty (e.g. confidence intervals) |
| ☒ | ☐ | For null hypothesis testing, the test statistic (e.g. *F*, *t*, *r*) with confidence intervals, effect sizes, degrees of freedom and *P* value noted <br> *Give P values as exact values whenever suitable.* |
| ☒ | ☐ | For Bayesian analysis, information on the choice of priors and Markov chain Monte Carlo settings |
| ☒ | ☐ | For hierarchical and complex designs, identification of the appropriate level for tests and full reporting of outcomes |
| ☒ | ☐ | Estimates of effect sizes (e.g. Cohen's *d*, Pearson's *r*), indicating how they were calculated |

*Our web collection on statistics for biologists contains articles on many of the points above.*

## Software and code

Policy information about availability of computer code

| Data collection | EPU (Thermo Fisher Scientific), an iBright FL1500 Imaging System (Thermo Fisher Scientific), Unicorn V7.1 |
|---|---|
| Data analysis | CryoSPARC v3.3, Coot v 0.9.4 , Phenix v 1.19.2-4158  USCF Chimera , USCF Chimera X, Pymolv1.8.2.0, Adobe Illustrator, Geneious prime (v2022.2), ImageJ (v1.53k), Clustal Omega, Jalview |

For manuscripts utilizing custom algorithms or software that are central to the research but not yet described in published literature, software must be made available to editors and reviewers. We strongly encourage code deposition in a community repository (e.g. GitHub). See the Nature Portfolio guidelines for submitting code & software for further information.

## Data

Policy information about availability of data

All manuscripts must include a data availability statement. This statement should provide the following information, where applicable:
- Accession codes, unique identifiers, or web links for publicly available datasets
- A description of any restrictions on data availability
- For clinical datasets or third party data, please ensure that the statement adheres to our policy

The raw image data are available from the EMPIAR database under access codes EMPIAR-11268. The reconstructed maps are available from the EMDB database under access codes EMDB-14847 and EMDB-14848. The atomic models are available in the PDB database, access codes PDB-ID 7ZOL and 7ZOQ.

## Human research participants

Policy information about studies involving human research participants and Sex and Gender in Research.

| | |
|---|---|
| Reporting on sex and gender | n/a |
| Population characteristics | n/a |
| Recruitment | n/a |
| Ethics oversight | n/a |

Note that full information on the approval of the study protocol must also be provided in the manuscript.

## Field-specific reporting

Please select the one below that is the best fit for your research. If you are not sure, read the appropriate sections before making your selection.

☒ Life sciences          ☐ Behavioural & social sciences          ☐ Ecological, evolutionary & environmental sciences

For a reference copy of the document with all sections, see nature.com/documents/nr-reporting-summary-flat.pdf

## Life sciences study design

All studies must disclose on these points even when the disclosure is negative.

| | |
|---|---|
| Sample size | Sample size for Cryo-EM data was determined by collecting data in the form of movies containing particle images to obtain a sufficient number of particles images that would provide high resolution 3D reconstruction following refinement based on the FSC threshold of 0.143. For the data of the Craspase (DmCas7-11-TPR-CHAT) complex, 18559 movies were collected. 65'755 particles was refined and yielded a cryo-EM map at 3.20 Å overall resolution and another set of 53'969 particles produced a cryo-EM map at an overall resolution of 3.03 Å based on FSC threshold of 0.143. Therefore, the number of particles obtained were sufficient to get high resolution structures sufficient to make the interpretations detailed in the manuscript.<br> For other experiments, no sample-size determination was performed, as the biochemical experiments followed standard practices. Replicates of all biochemical experiments was performed in order to ensure reproducibility of the results. |
| Data exclusions | Images suffering image drift, ice contamination, and/or cubic ice format were excluded during image processing. Particles in 2D classes showing no secondary structural features and in 3D classes showing unsatisfactory structural features were excluded from the final reconstructions in all datasets analyzed. |
| Replication | Cryo-EM data was obtained from 18559 observations in the form of movies from a single sample from a round of data collection. Biochemical asssays such as the nuclease activity assays were performed at least three independent times with similar results. EMSA assay was independently duplicated also with similar results. |
| Randomization | No randomization was involved as this study did not involve use of sample collection from different experimental groups involving participants |
| Blinding | Blinding is not applicable as this study did not involve sampling different experimental groups involving participants |

## Reporting for specific materials, systems and methods

We require information from authors about some types of materials, experimental systems and methods used in many studies. Here, indicate whether each material, system or method listed is relevant to your study. If you are not sure if a list item applies to your research, read the appropriate section before selecting a response.

### Materials & experimental systems

| n/a | Involved in the study |
|---|---|
| ☒ | ☐ Antibodies |
| ☒ | ☐ Eukaryotic cell lines |
| ☒ | ☐ Palaeontology and archaeology |
| ☒ | ☐ Animals and other organisms |
| ☒ | ☐ Clinical data |
| ☒ | ☐ Dual use research of concern |

### Methods

| n/a | Involved in the study |
|---|---|
| ☒ | ☐ ChIP-seq |
| ☒ | ☐ Flow cytometry |
| ☒ | ☐ MRI-based neuroimaging |

