## [Peer Review File · Nature Structural & Molecular Biology]

Peer Review Information

Manuscript Title: Structural insights into the regulation of Cas7-11 by TPR-CHAT

Corresponding author name(s): Henning Stahlberg, Dongchun Ni

Reviewer Comments & Decisions:

Decision Letter, initial version:
--

Message: 18th Aug 2022

Dear Dr. Stahlberg,

Thank you again for submitting your manuscript "Structural insights into the regulation of Cas7-11 by TPR-CHAT". I sincerely apologize for the delay in responding, which resulted from the difficulty in obtaining suitable referee reports. Nevertheless, we now have comments (below) from the 2 reviewers who evaluated your paper. In light of those reports, we remain interested in your study and would like to see your response to the comments of the referees, in the form of a revised manuscript.

You will see that all reviewers appreciate the results and find the conclusions timely and of wide interest. There are, however, several concerns and suggestions that should be addressed in a revision. Specifically, reviewer #2 asks for biochemical validation of nuclease inhibition by mutagenesis of the domains in TPR-CHAT shown to regulate this. We agree with the reviewer that this is an important point to be addressed (point #3) but can understand that further functional assays (point #4) are outside the scope of the present manuscript and would be willing to overrule on their requirement. Please be sure to address/respond to all concerns of the referees in full in a point-by-point response and highlight all changes in the revised manuscript text file. If you have comments that are intended for editors only, please include those in a separate cover letter.

We expect to see your revised manuscript within 6 weeks. If you cannot send it within this time, please contact us to discuss an extension; we would still consider your revision, provided that no similar work has been accepted for publication at NSMB or published elsewhere.

As you already know, we put great emphasis on ensuring that the methods and statistics

reported in our papers are correct and accurate. As such, if there are any changes that should be reported, please submit an updated version of the Reporting Summary along with your revision.

Reporting Summary:

Please note that all key data shown in the main figures as cropped gels or blots should be presented in uncropped form, with molecular weight markers. These data can be aggregated into a single supplementary figure item. While these data can be displayed in a relatively informal style, they must refer back to the relevant figures. These data should be submitted with the final revision, as source data, prior to acceptance, but you may want to start putting it together at this point.

Data availability: this journal strongly supports public availability of data. All data used in

accepted papers should be available via a public data repository, or alternatively, as Supplementary Information. If data can only be shared on request, please explain why in your Data Availability Statement, and also in the correspondence with your editor. Please note that for some data types, deposition in a public repository is mandatory - more information on our data deposition policies and available repositories can be found below: <https://www.nature.com/nature-research/editorial-policies/reporting-standards#availability-of-data>

[Redacted]

Sincerely,
Sara

Sara Osman, Ph.D.
Associate Editor
Nature Structural & Molecular Biology

Referee expertise:

Referee #1: CRISPR, cryo-EM

Referee #2: CRISPR, biochemistry and structural biology

Reviewers' Comments:

Reviewer #1:

Remarks to the Author:

The manuscript by Ekundayo et. al. presented the cryo-EM structures of Cas7-11 in complex with TPR-CHAT, revealing a mechanism of RNA guided RNA processing by this unique CRISPR-guided system. Cas7-11 is a novel CRISPR system that can target RNA specifically in a CRISPR RNA dependent manner. A caspase-like domain containing protein known as TPR-CHAT was shown to be an ancillary protein tightly bound to Cas7-11. The structures were determined to very high resolutions and the structural interpretation is quite convincing. There are a few minor points to be addressed before the manuscript being accepted for publication.

1. From structural analysis, the authors proposed that TRP-CHAT inhibits the activity of Cas7-11-crRNA. The authors should compare the activity of Cas7-11-crRNA alone and in complex with TRP-CHAT using the assay in Fig. 1b.

2. In line 102-103, the authors stated that the CTD of TPR-CHAT is a bona fide caspase-related peptidase simply based on structural analysis. I would suggest to tone down this statement as there is no biochemical evidence.

3. Label the domain boundary in Fig. 1a.

4. Extended data 10a, it would be much clearer to show the ribbon diagram of TPR-CHAT alone with three domains color coded. The figure legend for 10b should be "Topology diagram of DmTPR-CHAT CTD".

Reviewer #2:

Remarks to the Author:

This manuscript showed the novel structure of Cas7-11 complexed with TRP-CHAT, the Caspase complex around 3 Å resolution and proposed the model of Cas7-11 nuclease inhibition by TRP-CHAT. The Figures presenting the structures are good and the story is fascinating. This reviewer, however, request the following points for further considerations.

1. Cas7-11 is a nuclease targeting RNA substrate, cleaving two points. However, this manuscript contains a short introduction, which said nothing above about the basic activity of Cas7-11, which preclude broad readers to understand even at the first. Please describe the introduction more carefully.

2. The Cas7-11 structure solved by Cryo-EM was published in Cell the end of June, before submission of this manuscript. Therefore, the authors compare the structures more in detail about which is different between two structures using new figures.
3. Nevertheless, the present structure is a complex with TRP-CHAT, which is totally new, as compared to the previous paper. The nuclease inhibition by NTD as well as CTD of TRP-CHAT deserves appealing and the authors should present functional analysis that WTR and CTD mutants inhibit the nuclease activity of Cas7-11.
4. The middle caspase-like domain function is extremely important. The authors should perform some biochemical, proteome and cell-based functional analysis to elucidate the function of this caspase-like domain of TRP-CHAT. This should be one of the main issue of this manuscript as well as the physiological importance of Cas7-11.
5. Ext. Data Fig. 7 is lacking.

Once the authors address the above concerns by new experiments, I strongly recommend publication of this manuscript in Nature Structural & Molecular Biology.

Author Rebuttal to Initial comments

We thank the reviewers for their careful review of our manuscript and very useful comments and feedback. We have revised the manuscript accordingly and provide a point-by-point response in red text to the reviewer's comments below:

Reviewers' Comments:

Reviewer #1:

Remarks to the Author:

The manuscript by Ekundayo et. al. presented the cryo-EM structures of Cas7-11 in complex with TRP-CHAT, revealing a mechanism of RNA-guided RNA processing by this unique CRISPR-guided system. Cas7-11 is a novel CRISPR system that can target RNA specifically in a CRISPR RNA-dependent manner. A caspase-like domain containing protein known as TRP-CHAT was shown to be an ancillary protein tightly bound to Cas7-11. The structures were determined to very high resolutions and the structural interpretation is quite convincing. There are a few minor points to be addressed before the manuscript being accepted for publication.

1. From structural analysis, the authors proposed that TRP-CHAT inhibits the activity of Cas7-11-crRNA. The authors should compare the activity of Cas7-11-crRNA alone and in complex with TRP-CHAT using the assay in Fig. 1b.

As rightly noted by the reviewer, this is an important experiment to confirm our findings. We have used the nuclease activity assay as done in Figure 1a to compare the activity of Cas7-11-crRNA alone and in complex with TRP-CHAT. We purified full-length TRP-CHAT as shown in **Extended data 13a** and performed nuclease activity assays at varying concentrations of

DmTPR-CHAT and different time points in **Extended data 13b and c**. Our results clearly show that *DmTPR_CHAT* inhibits *DmCas7-11* target RNA cleavage activity and are also included in the text in **lines 134-137**.

2. In line 102-103, the authors stated that the CTD of *TPR-CHAT* is a bona fide caspase-related peptidase simply based on structural analysis. I would suggest to tone down this statement as there is no biochemical evidence.

We have now reworded this statement to: 'These findings show that the CTD harbours a putative caspase-related peptidase, with its substrate yet to be determined' as seen in **lines 106- 107**.

3. Label the domain boundary in Fig. 1a.

Figure 1a has been updated to include the amino acid positions of the domain boundaries of *Cas7-11* and *TPR-CHAT*

4. *Extended data 10a*, it would be much clearer to show the ribbon diagram of *TPR-CHAT* alone with three domains color coded. The figure legend for *10b* should be "Topology diagram of *DmTPR-CHAT* CTD".

Extended data 11a (previously *10a*) and the figure legend for *11b* (previously *10b*) have been updated accordingly

Reviewer #2:**Remarks to the Author:**

This manuscript showed the novel structure of Cas7-11 complexed with TRP-CHAT, the Craspase complex around 3 Å resolution and proposed the model of Cas7-11 nuclease inhibition by TRP-CHAT. The Figures presenting the structures are good and the story is fascinating. This reviewer, however, request the following points for further considerations.

1. Cas7-11 is a nuclease targeting RNA substrate, cleaving two points. However, this manuscript contains a short introduction, which said nothing above about the basic activity of Cas7-11, which preclude broad readers to understand even at the first. Please describe the introduction more carefully.

We thank the reviewer for highlighting this key point in the introduction. We have updated the introduction to include some detail on the activity of Cas7-11 in **lines 27-31**. Since our manuscript is a **brief communication**, a more detailed introduction is not feasible due to word count constraints.

2. The Cas7-11 structure solved by Cryo-EM was published in Cell the end of June, before submission of this manuscript. Therefore, the authors compare the structures more in detail about which is different between two structures using new figures.

Our manuscript was submitted to NSMB on the 27th of April, 2022 which is around two months before the Cell paper (Kato *et al.* 2022) was published. We have now performed structural comparisons with that, and also with another structure published in Science (Hu *et al.* 2022). These are shown in **Extended data 10**, with the differences in the Insertion domains and the position of the Cas11-like domain shown in **Extended data 10 b, c** and also mentioned in the main text in **lines 86-89**. Moreover, our paper describes the structure and regulation of Craspase from *D. magnum*, which is a different species from what has been published.

3. Nevertheless, the present structure is a complex with TRP-CHAT, which is totally new, as compared to the previous paper. The nuclease inhibition by NTD as well as CTD of TRP-CHAT deserves appealing and the authors should present functional analysis that WTR and CTD mutants inhibit the nuclease activity of Cas7-11.

We have performed target RNA cleavage and binding assays using full-length, deletion of CTD (CTD-del) and CTD alone TPR-CHAT constructs. The results in **Figure 3f and Extended data 13f** show that full-length TPR-CHAT and CTD-del inhibit Cas7-11 target RNA cleavage by inhibiting target binding to Cas7-11 but not the CTD alone construct. The results support the role of the NTD in inhibition via its association with the Insertion-finger of Cas7-11. We also performed target RNA cleavage assays with an Insertion-finger deletion (IF-del) in Cas7-11 and found that TPR-CHAT was no longer capable of inhibition as shown in **Extended data 3d**. These results strongly support the model for regulation, which was proposed based on our Craspase structure. These results are described in **lines 139-146**.

4. The middle caspase-like domain function is extremely important. The authors should perform some biochemical, proteome and cell-based functional analysis to elucidate the function of this caspase-like domain of TRP-CHAT. This should be one of the main issues of this manuscript as well as the physiological importance of Cas7-11.

We agree with the reviewer that the function of the caspase-like domain of TPR-CHAT is of importance. However, this is outside the scope of this study, in which we seek to provide insight into the mechanism of regulation of Cas7-11 by TPR-CHAT in the form of a **brief communication**. However, understanding the function of the caspase-like domain in *D. magnum* is important to us, but will have to be addressed as a separate study and separate manuscript.

5. Ext. Data Fig. 7 is lacking.

Ext. data Figure 7 was actually present on the same page as Ext. data figure 8. They will now be put on separate pages to avoid further confusion.

Once the authors address the above concerns by new experiments, I strongly recommend publication of this manuscript in *Nature Structural & Molecular Biology*.

Decision Letter, first revision:

Message: Our ref: NSMB-BC46295A

11th Oct 2022

Dear Dr. Stahlberg,

Thank you for submitting your revised manuscript "Structural insights into the regulation of Cas7-11 by TPR-CHAT" (NSMB-BC46295A). It has now been seen by the original referees and their comments are below. The reviewers find that the paper has improved in revision, and therefore we'll be happy in principle to publish it in *Nature Structural & Molecular Biology*, pending minor revisions to comply with our editorial and formatting guidelines.

We are now performing detailed checks on your paper and will send you a checklist detailing our editorial and formatting requirements in about two weeks. Please do not upload the final materials and make any revisions until you receive this additional information from us.

To facilitate our work at this stage, we would appreciate if you could send us the main text as a word file. Please make sure to copy the NSMB account (cc'ed above).

Thank you again for your interest in *Nature Structural & Molecular Biology*. Please do not hesitate to contact me if you have any questions.

Sincerely,

Sara

Sara Osman, Ph.D.
Associate Editor
Nature Structural & Molecular Biology

Reviewer #1 (Remarks to the Author):

The authors have fully addressed my questions and I have no further concerns. Considering the fierce competition in the field, I suggested that the manuscript should be accepted for publication without any further delay.

Reviewer #2 (Remarks to the Author):

The authors most completely addressed my concerns, while this reviewer is still interested in the function of caspase-like domain, which, the author claims, will be published in the next paper. Therefore, I now strongly recommend publication of this paper in Nat. Struct. Mol. Biol. without delay.

Decision Letter, author guidance:

Message: Our ref: NSMB-BC46295A

14th Oct 2022

Dear Dr. Stahlberg,

Thank you for your patience as we've prepared the guidelines for final submission of your Nature Structural & Molecular Biology manuscript, "Structural insights into the regulation of Cas7-11 by TPR-CHAT" (NSMB-BC46295A). Please carefully follow the step-by-step instructions provided in the attached file, and add a response in each row of the table to indicate the changes that you have made. Please also check and comment on any additional marked-up edits we have proposed within the text. Ensuring that each point is addressed will help to ensure that your revised manuscript can be swiftly handed over to our production team.

We would like to start working on your revised paper, with all of the requested files and forms, as soon as possible. If you can resubmit within the next week it is possible that your submission could be published before the end of 2022. Please get in contact with us if you anticipate any delays in resubmission.

If you have not done so already, please alert us to any related manuscripts from your group that are under consideration or in press at other journals, or are being written up

for submission to other journals (see: <https://www.nature.com/nature-research/editorial-policies/plagiarism#policy-on-duplicate-publication> for details).

In recognition of the time and expertise our reviewers provide to Nature Structural & Molecular Biology's editorial process, we would like to formally acknowledge their contribution to the external peer review of your manuscript entitled "Structural insights into the regulation of Cas7-11 by TPR-CHAT". For those reviewers who give their assent, we will be publishing their names alongside the published article.

Nature Structural & Molecular Biology offers a Transparent Peer Review option for new original research manuscripts submitted after December 1st, 2019. As part of this initiative, we encourage our authors to support increased transparency into the peer review process by agreeing to have the reviewer comments, author rebuttal letters, and editorial decision letters published as a Supplementary item. When you submit your final files please clearly state in your cover letter whether or not you would like to participate in this initiative. Please note that failure to state your preference will result in delays in accepting your manuscript for publication.

Cover suggestions

As you prepare your final files we encourage you to consider whether you have any images or illustrations that may be appropriate for use on the cover of Nature Structural & Molecular Biology.

Nature Structural & Molecular Biology has now transitioned to a unified Rights Collection system which will allow our Author Services team to quickly and easily collect the rights and permissions required to publish your work. Approximately 10 days after your paper is formally accepted, you will receive an email in providing you with a link to complete the grant of rights. If your paper is eligible for Open Access, our Author Services team will also be in touch regarding any additional information that may be required to arrange payment for your article.

Please note that *Nature Structural & Molecular Biology* is a Transformative Journal (TJ). Authors may publish their research with us through the traditional subscription access route or make their paper immediately open access through payment of an article-

processing charge (APC). Authors will not be required to make a final decision about access to their article until it has been accepted. [Find out more about Transformative Journals](https://www.springernature.com/gp/open-research/transformative-journals)

Authors may need to take specific actions to achieve [compliance](https://www.springernature.com/gp/open-research/funding/policy-compliance-faqs) with funder and institutional open access mandates. If your research is supported by a funder that requires immediate open access (e.g. according to [Plan S principles](https://www.springernature.com/gp/open-research/plan-s-compliance)) then you should select the gold OA route, and we will direct you to the compliant route where possible. For authors selecting the subscription publication route, the journal's standard licensing terms will need to be accepted, including [self-archiving policies](https://www.nature.com/nature-portfolio/editorial-policies/self-archiving-and-license-to-publish). Those licensing terms will supersede any other terms that the author or any third party may assert apply to any version of the manuscript.

Please use the following link for uploading these materials:
[Redacted]

Best regards,

Aimee Frier
Editorial Assistant
Nature Structural & Molecular Biology
nsmb@us.nature.com

On behalf of

Sara Osman, Ph.D.
Associate Editor
Nature Structural & Molecular Biology

Reviewer #1:
Remarks to the Author:

The authors have fully addressed my questions and I have no further concerns. Considering the fierce competition in the field, I suggested that the manuscript should be accepted for publication without any further delay.

Reviewer #2:

Remarks to the Author:

The authors most completely addressed my concerns, while this reviewer is still interested in the function of caspase-like domain, which, the author claims, will be published in the next paper. Therefore, I now strongly recommend publication of this paper in Nat. Struct. Mol. Biol. without delay.

Final Decision Letter:

Message: 4th Nov 2022

e:

Dear Dr. Stahlberg,

I am delighted to tell you that your manuscript NSMB-BC46295B has been accepted for publication in Nature Structural & Molecular Biology.

As discussed, due to the exceptional nature of your work, we will publish your paper on an accelerated schedule. **Please carefully review the details below and contact us immediately at nsmb@us.nature.com if you have any travel plans or other conflicts that may make you unable to respond to us for the next 5-7 days.**

In approximately 2 business days you will receive a link to choose the appropriate publishing options for your paper and complete the appropriate grant of rights necessary to publish your work. As it is vital that this process not be delayed, we strongly encourage you to <https://www.simpleminds.com/how-to-check-your-spam-filter-and-whitelist-emails/> whitelist the email address do-not-reply@springernature.com to ensure that this message is received.

Shortly after this step is completed, you will receive a link to your electronic proof via email with a request to make any necessary corrections as soon as possible. You will find that we have made minor changes to enhance the clarity of the text and to ensure that your paper conforms to the journal's style so we ask that you review these proofs carefully to ensure that we have not inadvertently introduced errors or altered the sense of your text in any way.

Please return your proof within 24 hours of receiving it. If you have any questions about your proofs or anticipate any delays please contact rjsproduction@springernature.com immediately.

Once again, you will not receive your proofs until the publishing agreement has been

received through our system and failure to respond will result in delays in moving forward with publishing your work.

Once a publication date is set for your paper, the Springer Nature press office will be in touch with the full embargo details. We request that you do not send out your own publicity or contact any journalists until you hear from us that the paper has a confirmed publication date.

If you would like to inform your Public Relations or Press Office about your paper, we suggest that you do so immediately to allow them as much time as possible to prepare an appropriate press release and organize publicity if they choose to do so. Please include your manuscript tracking number NSMB-BC46295B and the name of the journal, which they will need if they contact our press office.

Please note that Nature Structural & Molecular Biology is a Transformative Journal (TJ). Authors may publish their research with us through the traditional subscription access route or make their paper immediately open access through payment of an article-processing charge (APC). Authors will not be required to make a final decision about access to their article until it has been accepted. [Find out more about Transformative Journals](https://www.springernature.com/gp/open-research/transformative-journals)

If you have any questions about our publishing options, costs, Open Access requirements, or our legal forms, please contact ASJournals@springernature.com.

An online order form for reprints of your paper is available at https://www.nature.com/reprints/author-reprints.html. All co-authors, authors' institutions and authors' funding agencies can order reprints using the form appropriate to their geographical region.

Sincerely,
Sara

Sara Osman, Ph.D.
Associate Editor
Nature Structural & Molecular Biology

P.S. Click here if you would like to recommend Nature Structural & Molecular Biology to your librarian - this will link directly to the Recommend page.

<http://www.nature.com/subscriptions/recommend.html#forms>

** Visit the Springer Nature Editorial and Publishing website at www.springernature.com/editorial-and-publishing-jobs for more information about our career opportunities. If you have any questions please click here.**